# The AUTOTAC chemical biology platform for targeted protein degradation via the autophagy-lysosome system

Chang Hoon Ji[1,2,10], Hee Yeon Kim [1,2,10], Min Ju Lee[1,2,10], Ah Jung Heo[1,2], Daniel Youngjae Park[1], Sungsu Lim[3], Seulgi Shin[3,4], Srinivasrao Ganipisetti[5], Woo Seung Yang[3], Chang An Jung[2], Kun Young Kim[2], Eun Hye Jeong[2], Sun Ho Park[2], Su Bin Kim[1], Su Jin Lee[1], Jeong Eun Na[2], Ji In Kang [6], Hyung Min Chi[7], Hyun Tae Kim[2], Yun Kyung Kim [3,4✉], Bo Yeon Kim [6,8✉] & Yong Tae Kwon [1,2,9✉]

Targeted protein degradation allows targeting undruggable proteins for therapeutic applications as well as eliminating proteins of interest for research purposes. While several degraders that harness the proteasome or the lysosome have been developed, a technology that simultaneously degrades targets and accelerates cellular autophagic flux is still missing. In this study, we develop a general chemical tool and platform technology termed AUTOphagy-TArgeting Chimera (AUTOTAC), which employs bifunctional molecules composed of target-binding ligands linked to autophagy-targeting ligands. AUTOTACs bind the ZZ domain of the otherwise dormant autophagy receptor p62/Sequestosome-1/SQSTM1, which is activated into oligomeric bodies in complex with targets for their sequestration and degradation. We use AUTOTACs to degrade various oncoproteins and degradation-resistant aggregates in neurodegeneration at nanomolar $DC_{50}$ values in vitro and in vivo. AUTOTAC provides a platform for selective proteolysis in basic research and drug development.

[1] Cellular Degradation Biology Center and Department of Biomedical Sciences, College of Medicine, Seoul National University, Seoul 03080, Korea. [2] AUTOTAC Bio Inc., Changkkyunggung-ro 254, Jongno-gu, Seoul 03080, Korea. [3] Convergence Research Center for Brain Science, Brain Science Institute, Korea Institute of Science and Technology (KIST), Seoul 02792, Korea. [4] Division of Bio-Medical Science & Technology, KIST School, University of Science and Technology (UST), Seoul 02792, Korea. [5] Brown Cancer Center, University of Louisville, 529 S Jackson Street, Louisville, KY 40202, USA. [6] Anticancer Agents Research Center, Korea Research Institute of Bioscience and Biotechnology, Ochang, Cheongju 28116, Korea. [7] Department of Chemisty, Pohang University of Science and Technology, Pohang 37673, Korea. [8] Department of Biomolecular Science, KRIBB School, University of Science and Technology (UST), Daejeon 34113, Korea. [9] SNU Dementia Research Center, College of Medicine, Seoul National University, Seoul 110-799, Republic of Korea. [10] These authors contributed equally: Chang Hoon Ji, Hee Yeon Kim, Min Ju Lee. ✉email: yunkyungkim@kist.re.kr; bykim@kribb.re.kr; yok5@snu.ac.kr

In the central dogma, the genetic information in DNA is transcribed into RNA, which in turn is translated to protein. Recent advances in genetic bioengineering, exemplified by CRISPR, TALEN, and siRNA, have enabled selective destruction and functional silencing of DNA and RNA[1,2]. In contrast to the universally applicable nature of DNA and RNA editing, there are currently no general tools by which proteins are selectively recognized and targeted for destruction. Functional silencing of proteins via small molecule inhibitors, while highly penetrant, rapid, and straightforward, is limited to approximately only a fifth of the entire human proteome[3]. Such a technology, if available, will be readily applied as research tools and agents to selectively downregulate target proteins in cultured cells and in living organisms such as mice and flies. Selective degradation of disease-associated proteins also provides an attractive opportunity to develop degrader-type drugs.

Targeted protein degradation (TPD) is the latest of emerging modalities in drug discovery and development and typically employs heterobifunctional chimeric molecules comprised of a target binder linked to a degradation-inducing moiety, which offer an attractive therapeutic means to eradicate disease-associated proteins, especially those belonging to the once-considered undruggable proteome[3–7]. Current TPD technology, as exemplified by PROteolysis-TArgeting Chimera (PROTAC), is limited to inducing the ubiquitination of target substrate for its degradation[8]. Despite its advantages, however, ubiquitin (Ub)-dependent PROTAC is largely confined to a limited set of target proteins and of E3 ligases, due to technical difficulties in forming a substrate-PROTAC-E3 ternary complex[9]. These technical challenges, which to date have ruled out the possibility of a pan-ubiquitinating and promiscuous E3 ligase for PROTACs, limit the meaningful clinical targets of PROTAC to a handful of otherwise short-lived ubiquitinated substrates, especially oncoproteins[8,10,11]. Moreover, Ub-dependent PROTAC may not be an ideal TPD platform to degrade misfolded and aggregation-prone pathological protein species, such as those of neurodegenerative proteinopathies, as these protein species are typically resistant to unfolding and subsequent degradation by the proteasome due to size limitations[12,13]. Recent advances in the field of autophagy-based degraders or molecular glues, such as AUTAC, ATTEC, and LYTAC have been successful in degrading a number of targets via the lysosome[14–17]. However, LYTAC is applicable to only extracellular proteins, ATTEC directly targets mutant Htt or lipid droplets to the autophagosome without prior sequestration, and AUTAC utilizes S-guanylation that is still dependent on ubiquitination of the target[14–17]. These challenges of the existing TPD modalities necessitate further development of a generally applicable TPD platform independent of Ub and the proteasome.

Selective macroautophagy is a catabolic process by which unwanted or harmful cytoplasmic constituents including proteins, aggregates, and organelles are specifically sequestered by autophagosomes for lysosomal hydrolysis[18]. Selective targeting of autophagic cargoes calls for specific receptors such as p62 and other Sequestosome-like receptors (SLRs) that simultaneously recognize Ub chains on protein cargoes and the lipidated LC3 in the autophagosomal inner membrane via the UBA and the LIR domain, respectively[19]. While recent efforts—such as harnessing S-guanylation-inducing molecular tags (AUTAC) or targeting extracellular and secreted/membrane proteins directly to the lysosome via peptidic tags (LYTAC)—attempt to address the degradation of PROTAC-resistant cargoes, no degrader technology yet exists for directly sequestering and targeting proteins and their aggregates using autophagy cargo receptors (e.g., p62).

In the N-degron pathway, specific single N-terminal amino acids of proteins, termed N-degrons, are recognized by N-recognins to induce substrate degradation[20]. The arginylation branch of this pathway uses Arg, Lys, His (type 1), Phe, Tyr, Trp, Leu, and Ile (type 2) as N-terminal degrons, among which the Arg N-degron can be generated via ATE1 R-transferase-mediated conjugation of L-Arg to Asp, Glu, or oxidized Cys. Recently, we discovered that the Arg/N-degron pathway mediates not only UBR-dependent proteasomal clearance but also macroautophagic proteolysis, wherein the archetypal autophagic cargo adaptor p62/SQSTM1 acts as an N-recognin that binds type-1 and type-2 N-degrons via its ZZ domain[21]. Binding of the Nt-Arg residue through the p62-ZZ domain conformationally activates p62 into an autophagy-compatible form, accelerating its self-oligomerization, interaction with LC3, and autophagosome biogenesis, facilitating autophagic targeting of p62-cargo complexes in a multi-step manner[21–23].

Here, we developed a generally applicable degrader, AUTO-TAC, by which a broad range of cellular proteins can be selectively recognized and targeted to autophagic membranes for lysosomal degradation. Central to the mode of action in AUTOTAC is the ability of the p62-binding moiety to induce a conformational activation of otherwise inactive p62 into an autophagy-compatible form. Upon binding to the p62 ligand, p62 exposes PB1 and LIR domains, which respectively facilitates p62 self-polymerization in complex with targets and its interaction with LC3 on autophagic membranes. Thus, AUTOTACs can induce the degradation of a broad range of cellular proteins including their misfolded aggregates.

## Results

**Development of the AUTOTAC platform.** In contrast to the UPS in which each of the 800 E3 ligases possesses a specific clientele of substrates, selective autophagy employs only a handful of receptors for intracellular cargoes, amongst which p62 plays a dominant role. If a bifunctional molecule binds both a target and p62 and activate the otherwise inactive p62, any given proteins could be targeted for autophagic degradation in principle. We therefore developed autophagy-targeting ligands (ATL; p62-binding moieties) that bind and activate p62 into an autophagy-compatible form (Fig. 1a and Supplementary Fig. 1a–e). 3D structure modeling, followed by SAR (structure-activity relationship) studies, was employed on the ZZ domain of p62. Three compounds, YOK-2204, YOK-1304 and YTK-105 (Fig. 1b), showed high docking scores based on low/negative energy of the stable system (−5.8, −5.5, and −4.0 kcal/mol for YOK-1304, YOK-2204, and YTK-105, respectively) (Supplementary Fig. 2a–f) when modeled to the p62-ZZ domain. Specific residues of the ZZ domain critical for N-degron recognition, such as Phe168, Arg139, Ile127, Asp129, Asp147, and Asp149, were also identified (Supplementary Fig. 2a–f).

In vitro pulldown assays using biotinylated p62 ligands confirmed that p62 bound YT-8-8, YOK-2204, and YTK-105, as opposed to the negative control V-BiP-biotin peptide (Fig. 1c and Supplementary Fig. 1e). The efficacy of ATLs in inducing p62 polymerization, a critical step in cargo collection, was demonstrated using in vitro oligomerization assays (Fig. 1d). In co-immunostaining analyses, p62 ligands induced the formation and co-localization of not only p62 and LC3 punctate structures (Fig. 1e–g and Supplementary Fig. 2g), but also those of p62 and the lysosomal marker LAMP1 (Supplementary Fig. 2h, i). Next, we confirmed that p62-ZZ ligands accelerated the formation of p62-positive and WIPI2-positive (canonical marker of omega-somes and phagophores) punctate structures as early as 6 h (Supplementary Fig. 2l, m) and lasting at least 24 h (Supplementary Fig. 2j, k). These data suggest that ATLs not only activate and target p62 to autophagic membranes but also facilitate autophagosome biogenesis to receive incoming p62-cargo complexes for

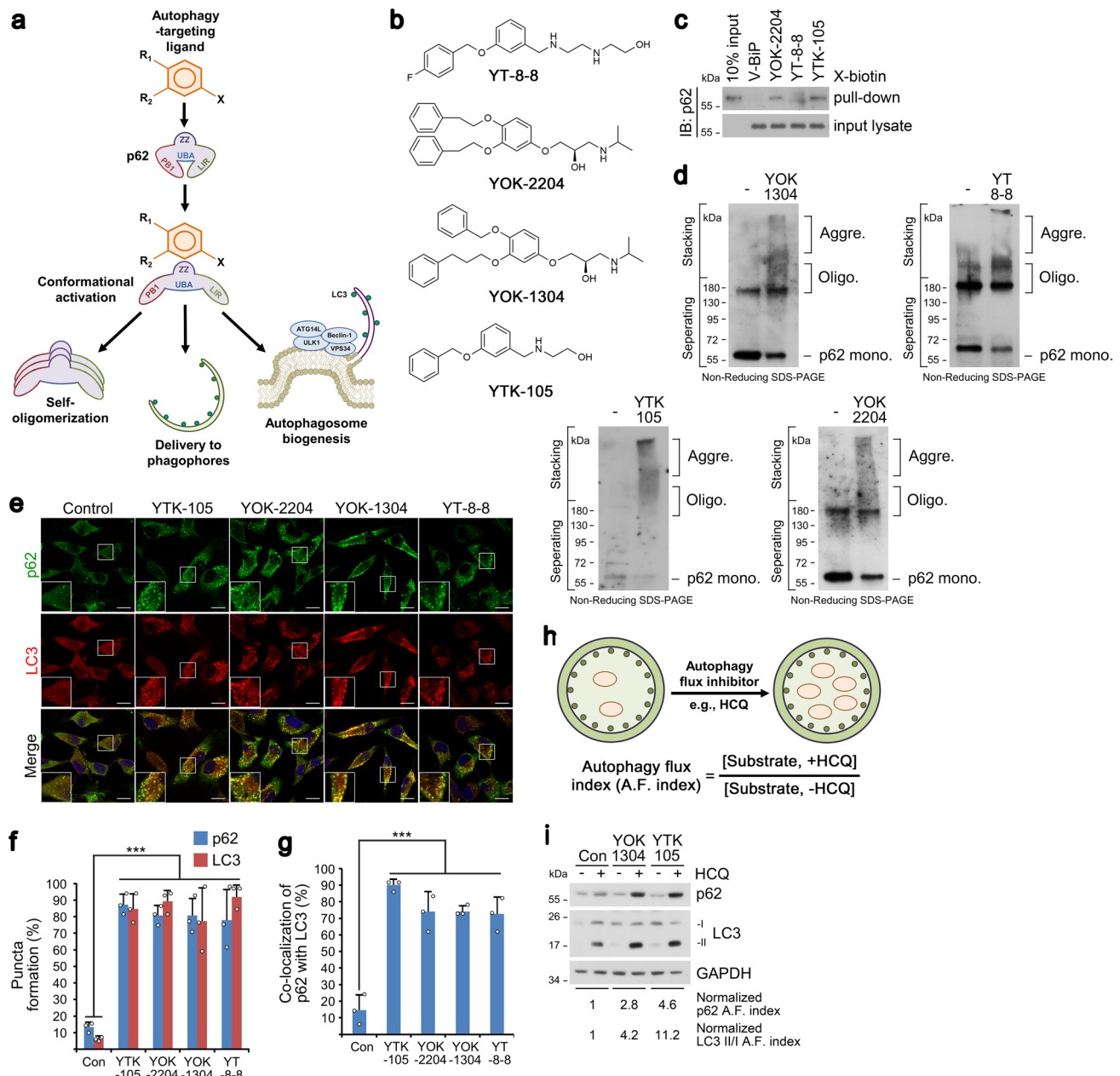

**Fig. 1 Nt-Arg-mimicking p62-ZZ ligands activate p62-dependent selective macroautophagy. a** A model illustrating the mode of action of autophagy-targeting ligands. **b** Chemical structures of autophagy-targeting ligands. **c** In vitro pulldown assay in HEK293T cells of the 12-mer V-BiP peptide or autophagy-targeting ligands. **d** In vitro p62 oligomerization assay in HEK293T cells incubated with the p62-ZZ ligands in (**b**). **e** ICC of HeLa cells treated with p62-ZZ ligands in (**b**) (2.5 μM, 24 h). Scale bar, 10 μm. **f, g** Quantification of (**e**) for puncta formation and co-localization, respectively (n = 3 biologically independent experiments each counting 50 cells). **h** A schematic/formulae for autophagy flux index. **i** Autophagic flux assay in HeLa cells treated with YOK-1304 or YTK-105 (2.5 μM, 24 h) in the presence or absence of HCQ (10 μM, 24 h). Data are presented as mean values ± SD where relevant. P-values (from a two-sided unpaired t test): ***$P < 0.000956$ (for p62 punctate formation), ***$P < 0.000539$ (for LC3 punctate formation), ***$P < 0.000925$ (for p62-LC3 co-localization). Source data are provided with this paper.

eventual lysosomal degradation. When autophagic flux indices were compared based on the ratio of substrate levels in the presence or absence of late-step autophagic inhibitors, p62-ZZ ligands enhanced autophagic turnover of p62 and LC3, indicative of increased autophagy flux (Fig. 1h, i and Supplementary Fig. 2n, o), in a proteasome-independent manner (Supplementary Fig. 2o). Importantly, the model p62 ligand YOK-2204 interacted with p62 but not NBR1, a very structurally and functionally similar autophagic cargo receptor also containing a ZZ domain (Supplementary Fig. 2p). Moreover, the interaction between YOK-2204 and mutant p62$^{D129A}$, carrying a point mutation in its

ZZ domain that is crucial for N-degron recognition, was severely crippled (Supplementary Fig. 2q). These results validate p62-ZZ ligands as ATLs capable of selectively interacting with and activating p62.

**Targeted degradation of soluble proteins by AUTOTAC.** We then used these p62-ZZ ligands to synthesize AUTOTACs, composed of target-binding ligands (TBLs) linked to p62-binding moieties via a repeating polyethylene glycol (PEG) moiety (Fig. 2a and Supplementary Fig. 3a–e). To test the degradative efficacy of

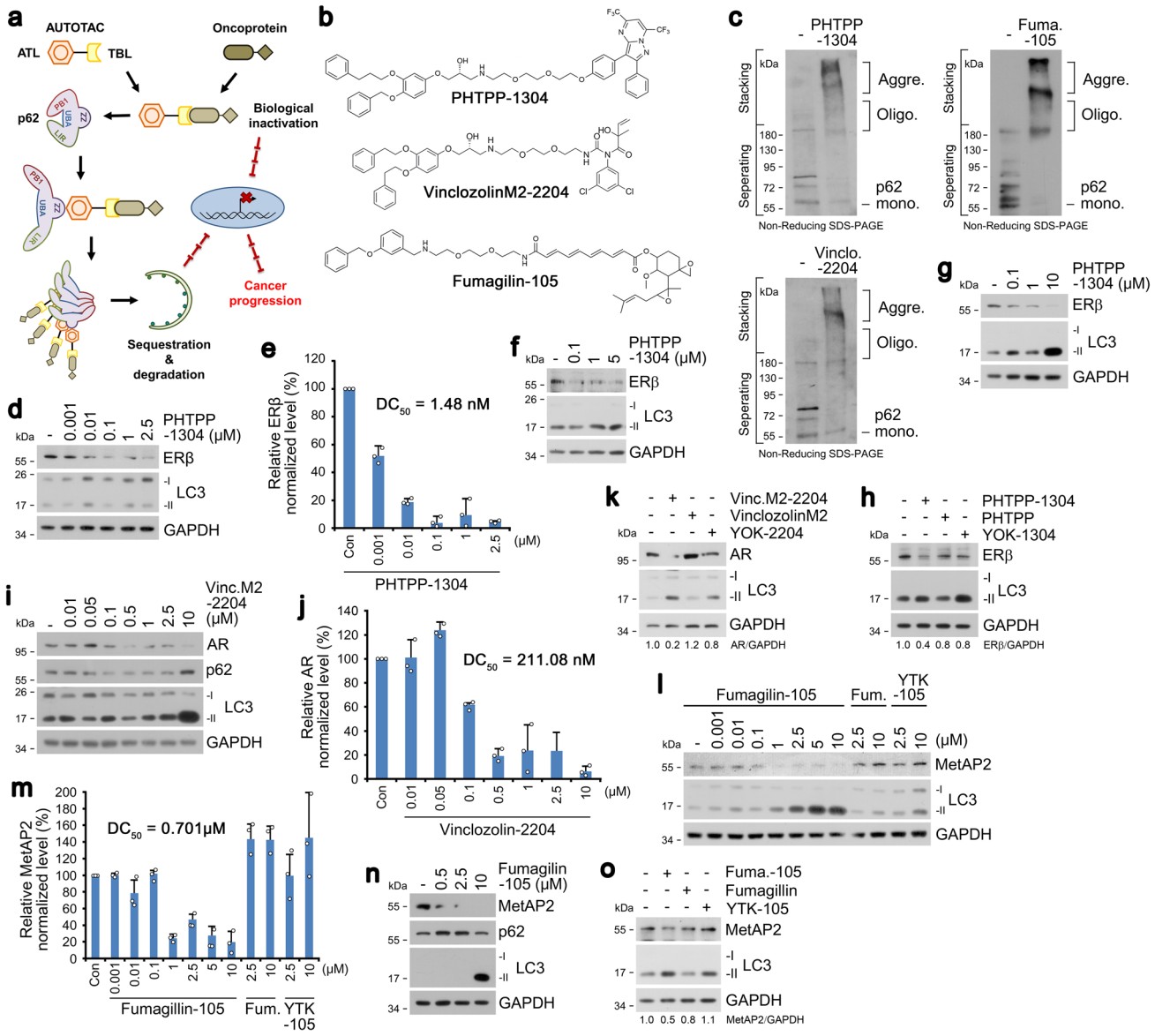

**Fig. 2 Targeted autophagic degradation of endogenous oncoproteins by AUTOTAC. a** A model illustrating oncoproteins-targeting AUTOTAC. **b** Chemical structures of oncoprotein-targeting AUTOTAC. **c** In vitro p62 oligomerization assay in HEK293T cells incubated with PHTPP-1304, VinclozolinM2-2204, or Fumagillin-105. **d** Western blot (WB) in HEK293T cells treated with PHTPP-1304 at the indicated concentrations (24 h). **e** Densitometry of (**d**) ($n = 3$). **f, g** WB in ACHN and MCF7 cells, respectively, treated with PHTPP-1304 at the indicated concentrations (24 h). **h** WB in MCF7 cells treated with PHTPP-1304, PHTPP or YOK-1304 (1 μM, 24 h). **i** WB in LNCaP cells treated with VinclozolinM2-2204 at the indicated concentrations. **j** Densitometry of (**i**) ($n = 3$). **k** WB in LNCaP cells treated with VinclozolinM2-2204, Vinclozolin or YOK-2204 (1 μM, 24 h). **l** WB in HEK293T cells treated with Fumagillin-105, Fumagillin or YTK-105 at the indicated concentrations (24 h). **m** Densitometry of (**l**) ($n = 3$). **n** WB in U87-MG cells treated with Fumagillin-105 at the indicated concentrations (24 h). **o** WB in U87-MG cells treated with Fumagilin-105, Fumagilin or YTK-105 (1 μM, 24 h). When indicated, $n = 3$ biologically independent experiments. Data are presented as mean values ± SD where relevant. Source data are provided with this paper.

AUTOTACs, we sought out to degrade estrogen receptor beta (ERβ) using its nonsteroidal and synthetic ligand, PHTPP (Fig. 2a, b). PHTPP-based AUTOTAC (PHTPP-1304) induced self-oligomerization of p62 (Fig. 2c) and degradation of ERβ at half-degrading concentration values ($DC_{50}$) of ~2 nM in HEK293T cells (Fig. 2d, e) and <100 nM in ACHN renal carcinoma and MCF-7 breast cancer cells (Fig. 2f, g). Sustained degradation and maximal clearance at the 24 h mark ($D_{max, 24 hr}$) values were observed at 10–100 nM (Fig. 2d, e). In contrast, no significant degradation was achieved with counterpart ATL and TBL (Fig. 2h). Next, we sought out to degrade androgen receptor (AR) and methionine aminopeptidase 2 (MetAP2) using

vinclozolinM2, a metabolite of vinclozolin, and fumagillol, a hydrolyzed product of fumagillin (Fig. 2b). Fumagillin covalently binds MetAP2 via a spiro-epoxide moiety[24], and vinclozolin derivatives and metabolites inhibit androgen binding to AR[25,26]. As expected, AR and MetAP2 AUTOTACs efficiently induced p62 self-polymerization (Fig. 2c). Moreover, vinclozolinM2-2204 exhibited $DC_{50}$ of ~200 nM for AR in LNCaP prostate cancer cells (Fig. 2i, j). Fumagillin-105 exhibited $DC_{50}$ of ~0.7 μM for MetAP2 in HEK293 cells with $D_{max, 24 hr}$ of ~1–10 μM (Fig. 2l, m) and $DC_{50}$ of ~500 nM in U87-MG glioblastoma cells (Fig. 2n). Such degradative efficacy was not observed with their ATLs or TBLs for both AR- and MetAP2-targeting AUTOTACs (Fig. 2k,

o). These results validate AUTOTAC as robust degraders for targeted proteolysis of intracellular oncoproteins.

Co-localization analysis in LNCaP cells showed that vinclozolinM2-2204 induced the formation of AR⁺LC3⁺ autophagic membranes (Fig. 3a, b). PHTPP-1304 in ACHN cells facilitated dosage-dependent formation of p62⁺ERβ⁺ puncta subject to autophagic flux when treated with the late-step autophagic flux inhibitor hydroxychloroquine (Fig. 3c, d). Additionally, PHTPP-1304 treatment induced the punctate formation and co-localization of ERβ and the omegasome marker WIPI2-GFP (Supplementary Fig. 4a). Consistently, fumagillin-105 treatment drastically up-regulated the autophagic flux of MetAP2, leading to lysosomal degradation via autophagosomes (Fig. 3e). Crucially, the degradation of ERβ by PHTPP-1304 was completely abolished by RNA interference of either *p62* or *ATG5* (Fig. 3f), which were corroborated by immunostaining analyses of ERβ or MetAP2 puncta formation present only in wild-type but not *p62*⁻/⁻ or *ATG5*⁻/⁻ mouse embryonic fibroblasts (Supplementary Fig. 4b–d). These results demonstrate that AUTOTAC drives lysosomal degradation of target proteins via p62-dependent macroautophagy.

As an autophagic cargo adaptor, p62 contains a LIR domain for LC3 interaction on autophagic membranes and a UBA domain that binds poly-Ub chains (Supplementary Fig. 4e). Thus, we determined whether AUTOTAC-based proteolysis depends on ubiquitination of target substrate and recognition of Ub chains via the p62-UBA domain. Importantly, AUTOTAC-driven degradation of MetAP2, ERβ, and AR remained not only intact but was even enhanced under *Ubb* interference (Fig. 3g, h and Supplementary Fig. 4f), possibly as a consequence of compensatory crosstalk between the UPS and autophagy. Following this vein of reasoning, we speculated that AUTOTAC-mediated degradation would be even more apparent in conditions of proteasome impairment. Indeed, degradation of ERβ upon PHTPP-1304 treatment with proteasomal inhibition in MCF7 cells exhibited subnanomolar $DC_{50}$ and $D_{max}$ values (Supplementary Fig. 4g). Moreover, this robust degradation persisted up to at least 8 h post-washout of PHTPP-1304 (Supplementary Fig. 4h), suggesting that AUTOTACs may display sustained degradative efficacy and be recycled from the lysosome. Taken together, these results suggest that AUTOTAC does not require ubiquitin-dependent and PPI-driven cooperativity for its sustained autophagic proteolysis.

**Therapeutic efficacy of AUTOTAC in cancer signaling**. To assess the therapeutic efficacy of autophagy-targeting degraders, we compared the downstream signaling of AR and ERβ in cells treated with either AUTOTACs or their cognate TBLs. Dihydrotestosterone (DHT) and estradiol (E2) are natural agonists that respectively bind AR and ERβ and induce their dimerization and nuclear translocation, resulting in transcriptional activation of downstream proteins[27,28]. When the levels of EGFR and p-Akt/Akt were measured in DHT-activated cells, vinclozolinM2-2204 inhibited AR pathways approximately 4-folds more efficiently than its TBL (Fig. 3i). Similarly, PHTPP-1304 inhibited ERβ signaling 10-folds more efficiently than its TBL as determined by the levels of EGFR, p-ERK/ERK and p-Akt/Akt in E2-stimulated LNCaP cells (Fig. 3j). Next, we examined the potency of AUTOTACs in cancer cell growth and progression. WST-based viability assays in ACHN cells showed that PHTPP-1304 exerted a ~5-fold higher cytotoxicity ($IC_{50}$, 3.3 μM) than those of its p62-binding ligand (>20 μM for YOK-1304) or TBL (18 μM for PHTPP) (Fig. 3m). VinclozolinM2-2204 also exhibited higher cytotoxicity ($IC_{50}$, 4.7 μM) as compared with its p62 ligand (>100 μM) and TBL (>100 μM) (Fig. 3n). When the efficacy of

degraders was assessed using wound healing assays, PHTPP-1304 more efficiently inhibited cell migration compared with p62-binding moiety or TBL (Supplementary Fig. 4i, j). Analogous assays targeting MetAP2 revealed that fumagillin-105 more efficiently inhibited the migration of U87-MG glioblastoma cells as early as 4 h and up to 24 h post-scratch (Fig. 3k, l). Finally, flow cytometry showed that prolonged exposure to fumagillin-105 induced programmed cell death as marked by the sub-G1 subpopulation (Supplementary Fig. 4k). These data demonstrate the therapeutic advantage of AUTOTACs against cancer cell growth and progression.

**AUTOTAC enables the targeting of Ub-conjugated misfolded protein aggregates to the lysosome**. Most proteins are misfolded or damaged at least once during their limited lifespans and thus necessitate their turnover via the UPS or autophagy. Soluble misfolded proteins are primarily degraded through the UPS, which involves unfolding into nascent polypeptides and cleavage by the proteasome[12,29,30]. However, the proteasome has an inner diameter as narrow as 13 Å whose pore is inaccessible to oligomers and aggregates and clogged by partially misfolded substrates, leaving autophagy as possibly the last line of defense against pathogenic aggregates[31,32]. We therefore applied AUTOTAC for UPS-resistant misfolded proteins and their oligomeric/aggregated species (Fig. 4a).

We first searched for a chemical chaperone that selectively recognizes the exposed hydrophobic regions as a universal signature of misfolded proteins. Screening of various compounds identified 4-phenylbutyric acid (PBA), an FDA-approved drug and chemical chaperone that improves proteostasis and ameliorates misfolding-induced ER stress[33]. PBA-1105 and PBA-1106 AUTOTACs (Fig. 4b) efficiently activated p62 and triggered its self-oligomerization (Fig. 4c). Immunoblotting analyses showed that PBA-1105 increased the autophagic flux of Ub-conjugated aggregates under prolonged proteasomal inhibition (Fig. 4d, flux indices 1 vs. 2.2), which was increasingly sustained for at least 48 h (Supplementary Fig. 5a). When their intracellular distribution was visualized using immunofluorescence analyses, PBA-1106 facilitated the formation of Ub⁺ cytosolic puncta, the vast majority of which colocalized with p62⁺ puncta (Fig. 4f, g) as well as p62⁺LC3⁺ autophagic membranes (Supplementary Fig. 5b). Specifically, while PBA-1106 treatment alone compared to control seemingly did not affect the number of p62 punctate structures, the increase in p62 punctate structures was drastically apparent in conditions of autophagy inhibition (14 ± 4.2 vs. 6 ± 2.4 p62 puncta structures). Strikingly, no such efficacy was observed with PBA or p62-binding ligand alone (Fig. 4d, f, g and Supplementary Fig. 5b). Moreover, siRNA-mediated knockdown of either *p62* or *ATG7* abolished PBA AUTOTAC-dependent degradation of Ub-conjugated protein aggregates (Supplementary Fig. 5c).

In neurodegeneration and other proteinopathies, aggregation-prone misfolded proteins inherently form oligomers that aggregate into fibrillary species. The screening of TBLs that selectively recognize the oligomeric signature of proteins and their aggregates yielded Anle138b, a phase 1 clinical trial compound that binds oligomers and aggregates of neurodegenerative proteinopathies[34,35]. Anle138b-F105 facilitated p62 self-polymerization (Fig. 4c) and autophagic flux of Ub-conjugated oligomeric/aggregated proteins (Fig. 4e) via p62-associated macroautophagy (Fig. 4h, i). In the same vein, Anle138b-F105 treatment induced the punctate formation and co-localization of Ub-conjugated protein aggregates with the omegasome marker WIPI2-GFP (Supplementary Fig. 5d) and the lysosome marker LAMP1 (Supplementary Fig. 5e). Similar to PBA-based

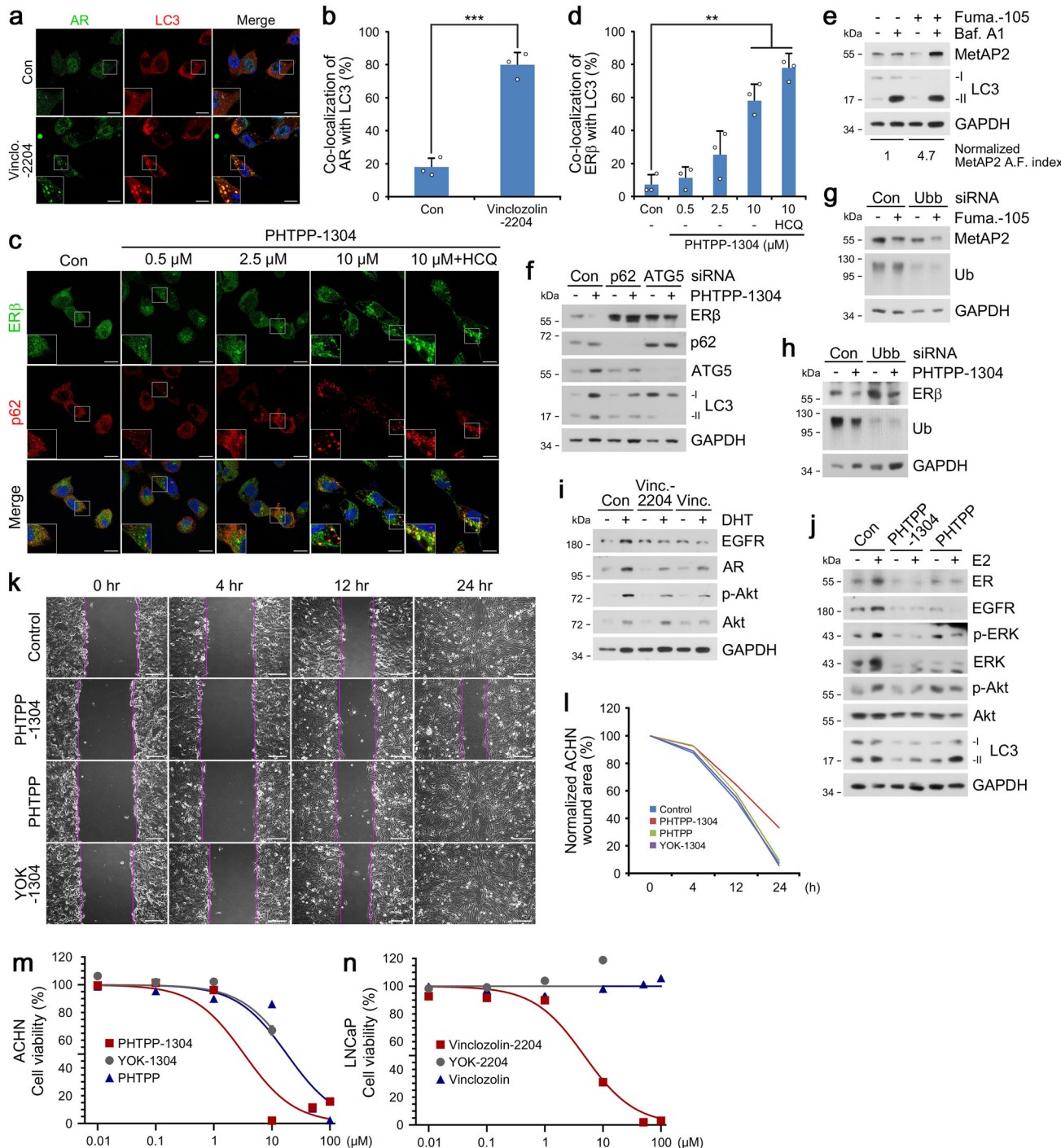

**Fig. 3 Targeted degradation using AUTOTAC inactivates oncogenic signaling. a** ICC of LNCaP cells treated with Vinclo.-2204 (2.5 µM, 24 h). Scale bar, 10 µm. **b** Quantification of (**a**) (*n* = 3 biologically independent experiments each counting 50 cells). **c** ICC of ACHN cells treated with PHTPP-1304 at the indicated concentrations and HCQ (10 µM) (24 h). Scale bar, 10 µm. **d** Quantification of (**c**) (*n* = 3 biologically independent experiments each counting 50 cells). **e** WB in U-87 MG cells treated with fumagillin-105 (1 µM, 24 h) with or without bafilomycin A1 (200 nM, 6 h). **f** WB in HEK293T cells treated with PHTPP-1304 (0.1 µM, 24 h) under siRNA-mediated knockdown of *p62* and *ATG5* (40 nM, 48 h). **g** WB in HeLa cells treated with fumagillin-105 (0.1 µM, 24 h) following RNA interference of *Ubb* (40 nM, 48 h). **h** Identical to (**g**), but with PHTPP-1304 for ERβ. **i** WB in LNCaP cells treated with vinclozolinM2-2204 (2.5 µM) or inclozolin (10 µM) with or without DHT (15 nM) (24 h). **j** Identical to (**i**) but with PHTPP-1304 (0.5 µM) or PHTPP (5 µM) with or without E2 (10 nM) (24 h). **k** Wound healing assay in ACHN cells treated with PHTPP-1304, PHTPP, or YOK-1304 (5 µM) at the indicated time points. **l** Quantification of (**k**) (*n* = 2 biologically independent experiments). Scale bar, 100 µm. **m**, **n** Cell viability assay of ACHN and LNCaP cells treated with the indicated compounds and concentrations (*n* = 2 biologically independent experiments). Data are presented as mean values ± SD where relevant. *P*-values (from a two-sided unpaired *t* test): \*\*\**P* < 0.000276, \*\**P* < 0.00161. Source data are provided with this paper.

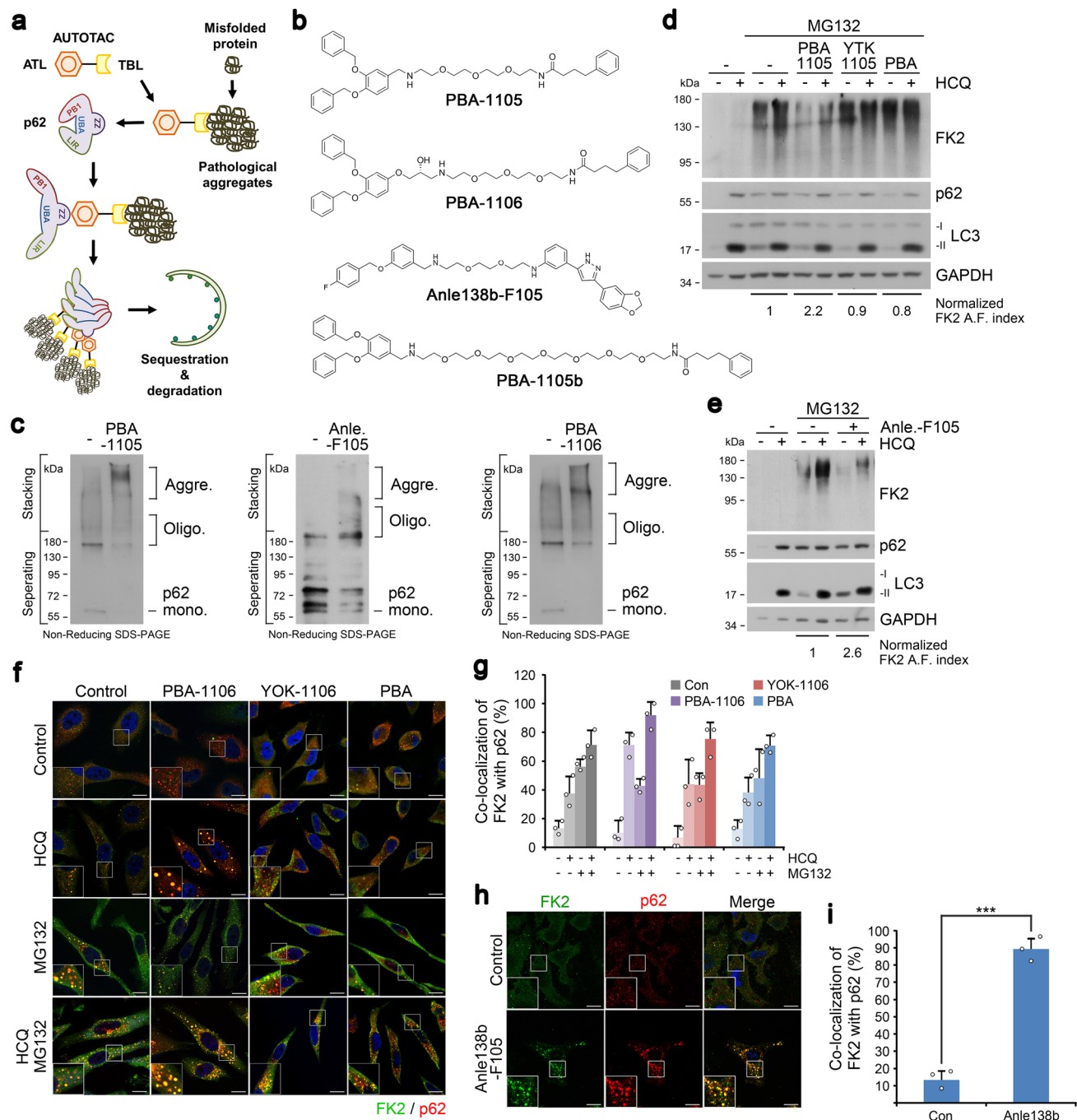

**Fig. 4 Targeted autophagic delivery and degradation of misfolded protein cargoes using AUTOTACs with aggregate-binding warhead. a** A model illustrating the mode of action of aggregate-targeting AUTOTAC. **b** Chemical structures of PBA-1105, PBA-1106, Anle138b-F105 and PBA-1105b. **c** In vitro p62 oligomerization assay in HEK293T cells incubated with the indicated compounds in (**b**). **d** WB in HEK293T cells treated with MG132 (1 μM, 24 h), HCQ (10 μM, 24 h), PBA-1105, YTK-1105, and PBA (1 μM, 24 h). **e** Identical to (**d**), but with Anle-F105. **f** ICC of HeLa cells treated with HCQ (10 μM, 24 h), MG132 (2 μM, 18 h) or both in the presence of PBA-1106, YOK-1106 or PBA (1 μM, 24 h). Scale bar, 10 μm. **g** Quantification of (**f**) (n = 3, 50 cells). **h** ICC of HeLa cells treated with Anle138b-F105 (1 μM, 24 h). Scale bar, 10 μm. **i** Quantification of (**h**) (n = 3, 50 cells). When indicated, n = 3 biologically independent experiments each counting 50 cells. Data are presented as mean values ± SD where relevant. P-values (from a two-sided unpaired t test): ***P < 7.66E−05. Source data are provided with this paper.

AUTOTAC, autophagic proteolysis of Ub-conjugated protein aggregates by Anle138b-F105 treatment was nullified by *p62* or *ATG7* interference (Supplementary Fig. 5f).

Desminopathies are systemic disorders caused by dysfunctional mutations in desmin or alphaB-crystallin, which cripple the intracellular filamentous network in cardiac and skeletal muscle cells and eventually induce muscle weakness, including cardiac and respiratory failure[36]. The many mutant and pathological protein species attributed to this disorder share a common trait in that they misfold, aggregate, and accumulate along with ubiquitin and other amyloidogenic proteins into insoluble granulo-filamentous material[36]. While wild-type desmin is not known to aggregate whatsoever, aggregation and accumulation of mutant desmin (which prevents its normal turnover via the UPS) is

known to disrupt protein homeostasis, including but not limited to chaperone deficiency, proteasome impairment, and mitochondrial dysfunction[36]. Thus, to generalize the efficacy of the AUTOTAC platform to misfolded protein aggregates, we tested the degradative efficacy of PBA-1105 and Anle138b-F105 against wild-type desmin and mutant, aggregation-prone desminL385P. Notably, all the tested AUTOTACs (i.e., PBA-1105, PBA-1105b, and Anle138b-F105) induced the degradation of exogenously expressed mutant desminL385P but not wild-type desmin in a concentration- (Supplementary Fig. 5g, h) and macroautophagy-dependent (Supplementary Fig. 5i) manner. Collectively, our data validate AUTOTAC-facilitated autophagic proteolysis of otherwise non-degradable, Ub-conjugated and UPS-resistant oligomeric or aggregated proteins by recognizing their exposed hydrophobic motifs or oligomeric signature.

**Degradation of pathological aggregates of neurodegenerative diseases using AUTOTAC.** Neurodegenerative diseases such as Alzheimer's disease are associated with an ever-increasing accumulation of degradation-resistant misfolded hallmark protein aggregates. Traditional approaches for developing ligands that alter the activity of neurodegeneration-associated targets are not applicable for pathological aggregates, leaving degraders as possibly the only therapeutic means. Since PROTAC-based approaches are inherently incapable of degrading large oligomers and aggregates, we tested whether AUTOTACs can target aggregation-prone P301L tau mutant that forms neurofibrillary tangles because of its prion-like seeding behavior[37].

In vitro pulldown assays confirmed that the chemical chaperone 4-PBA bound tauP301L stably expressed in SH-SY5Y cells (Supplementary Fig. 6a). PBA-1105 and PBA-1106 AUTOTACs induced autophagic degradation of stably expressed mutant tau at $DC_{50}$ of ~1–10 nM and $D_{max, 24 hr}$ of 100 nM, followed by a hook effect at higher concentrations (Fig. 5a–d and Supplementary Fig. 6b). In contrast to PBA AUTOTACs, virtually no degradation was observed with their TBL or p62-ZZ ligands (Fig. 5a–d and Supplementary Fig. 6b). We next confirmed that AUTOTAC-mediated degradation may not be critically dependent on linker length by synthesizing and confirming the anti-tauP301L degradative efficacy of PBA-1105b, which carries a drastically longer PEG-based linker than PBA-1105 (Fig. 4b and Supplementary Fig. 6c). Similar to PBA-based degraders, Anle138b-F105 targeted tauP301L for lysosomal degradation at $DC_{50}$ of ~3 nM, as opposed to its TBL or p62 ligand (Fig. 5e–h). Autophagy-based targeted degradation was obvious as early as 30 min and reached a sustained maximal effect from 3 hrs onwards (Fig. 5i). Anle138b-F105-mediated degradation of mutant tau species persisted up to at least 8 h post-washout (Supplementary Fig. 6d). These data raise the likelihood that AUTOTACs are linker length-insensitive and exhibit sustained efficacy against tau oligomers and aggregates.

We next confirmed whether AUTOTACs exert their efficacy by directly targeting oligomeric and aggregated species. Co-localization analysis showed that PBA-1105 AUTOTAC selectively induced the sequestration and autophagic targeting of tauP301L inclusion bodies in contrast to YTK-1105 or PBA (Fig. 5j, k). Consistently, in vivo aggregation assays revealed that PBA-1105 effectively eliminated high-molecular weight tau aggregates (Fig. 5l). Similarly, detergent-based fractionation of tauP301L into insoluble or soluble species revealed that only PBA-1105, in contrast to its TBL or p62-binding moiety, promoted autophagic degradation of not only detergent-soluble but critically also -insoluble species (Supplementary Fig. 6e, f). Next, we used the phosphatase inhibitor okadaic acid to examine the efficacy of PBA-based degraders against hyperphosphorylated

tau species, which act as a seedbed for tau aggregation. While okadaic acid treatment impaired the autophagic flux of mutant tauP301L (Supplementary Fig. 6g, lanes 1 and 2 vs. 5 and 6), presumably due to hyper-sequestration of tau, PBA-1106 significantly rescued autophagic flux of these otherwise non-digestible species (Supplementary Fig. 6g, lanes 3 and 4 vs. 6 and 8). These results were consistent with our observation that PBA-1105 treatment led to an increase in the number of GFP-quenched RFP-GFP-hTauP301L pre-formed inclusion bodies following their hyperphosphorylation and aggregation via prior treatment with okadaic acid, signaling their lysosomal digestion (Supplementary Fig. 6h, i). Detergent-based insoluble/soluble fractionation of okadaic acid-induced hyperphosphorylated tau further confirmed that PBA-1105 not only rescued but drastically accelerated the autophagic flux of both phosphorylated and total tau species (Fig. 5m, autophagy flux indices). Importantly, co-immunoprecipitation analyses revealed that Anle138b-F105 treatment induced not only the degradation of tauP301L but also its interaction with mutant p62 lacking the UBA domain, which is normally required for the interaction (Supplementary Fig. 6j). These data suggest that AUTOTAC provides a platform to target aggregation-prone misfolded proteins in neurodegenerative proteinopathies for lysosomal degradation in an ubiquitin-independent manner.

Next, we tested whether AUTOTAC is applicable for other neurodegeneration-associated proteins such as mutant huntingtin. HeLa cells were engineered to stably express wild-type (Q25) or mutant (Q97) huntingtin based on their nuclear localization/export signals (Htt-NLS-GFP or Htt-NES-GFP). Autophagy flux assays showed that PBA AUTOTACs induced lysosomal degradation of both nucleus- and cytosol-resident mutant huntingtin (Htt-NES-Q97 and Htt-NLS-Q97) at $DC_{50}$ of 0.1–1 µM (Supplementary Fig. 7a, b). Similar degradation efficacy was observed with transiently expressed mutant huntingtin, HDQ103 (Supplementary Fig. 7c). PBA AUTOTACs exhibited no significant degradative efficacy against wild-type huntingtin, HttQ25 (Supplementary Fig. 7d, e), further supporting their specificity to mutant Htt. As expected, their p62-binding ligands exhibited little efficacy against neither wild-type nor mutant Htt proteins (Supplementary Fig. 6b, e). Similar to PBA-based degraders, Anle138b-F105 also showed $DC_{50}$ of ~3 nM against the nuclear subpopulation of Htt (HttQ97-NLS-GFP) 24 h post-treatment (Supplementary Fig. 7f–i). When visualized using immunostaining analyses, Anle138b-F105 selectively promoted autophagic targeting of mutant HttQ103 as determined by co-localization of HttQ103 and LC3 in cells treated with hydroxy-chloroquine (Supplementary Fig. 7j, k). Given that nuclear LC3 does not localize to autophagic membranes unless it is retro-translocated and post-translationally modified in the cytosol, AUTOTACs likely bind and activate nuclear p62 (or cytosolic p62 that translocates to the nucleus) and deliver TBL-bound mutant HttQ97 to the cytosol for proteolysis. These data demonstrate that AUTOTAC is generally applicable for a broad range of pathogenic aggregates in neurodegeneration.

**AUTOTAC mediates the eradication of tau aggregates from mouse brains.** Increasing evidence points to soluble tau oligomers as causative agents in tau pathology due to their neurotoxic effect and a prion seed-like behavior for self-propagation[38]. To date, there are no general methods for targeted degradation of pathological protein oligomers and aggregates in neurodegeneration. We have previously developed hTauP301L-BiFC transgenic mice that express human tauP301L in the brain by employing bimolecular fluorescence complementation to visualize soluble tau oligomers[39] (Fig. 6a). We assessed the efficacy of PBA-1105 AUTOTAC, which showed

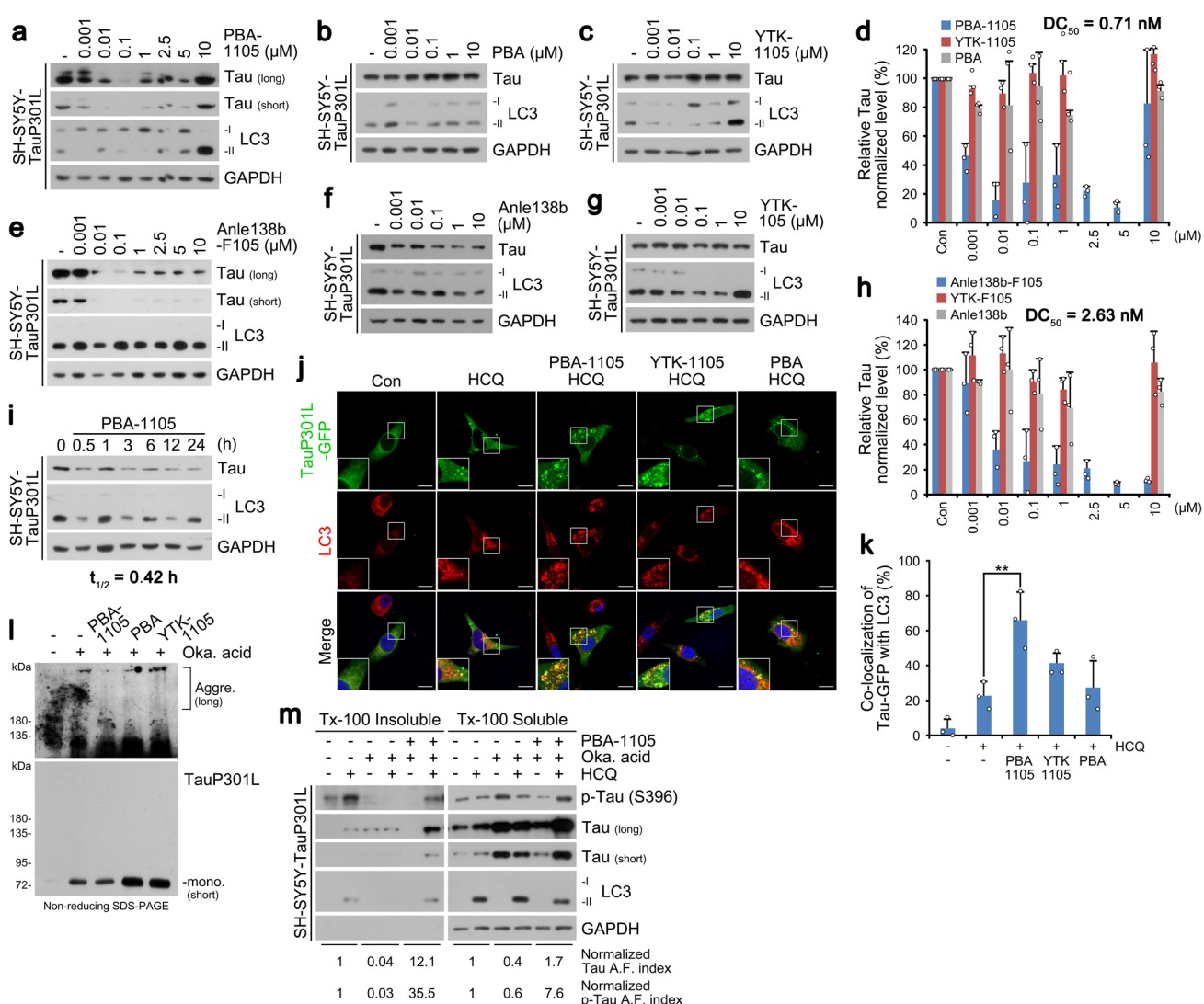

**Fig. 5 Selective degradation of pathological aggregation-prone tau species by aggregate-binding AUTOTAC. a–c** WB in SH-SY5Y-tauP301L cells treated with PBA-1105, PBA, or YTK-1105 at the indicated concentrations. **d** Densitometry of (**a**, **b**, and **c**) (n = 3 biologically independent experiments). **e–g** Same as (**a–c**) but with Anle138b-F105, Anle138b, or YTK-105. (**h**) Densitometry of (**e**, **f**, and **g**) (n = 3 biologically independent experiments). **i** WB in SH-SY5Y-tauP301L cells treated with PBA-1105 (0.1 μM) at the indicated time points. **j** ICC of HeLa cells expressing recombinant TauP301L-GFP and treated with the indicated compounds (1 μM, 24 h) and HCQ (10 μM, 24 h). Scale bar, 10 μm. **k** Quantification of (**j**) (n = 3 biologically independent experiments each counting 50 cells). **l** In vivo oligomerization assay in SH-SY5Y-tauP301L cells treated with okadaic acid (15 nM, 24 h) and the indicated compounds (0.1 μM, 24 h). **m** Triton X-100-fractionation assay in SH-SY5Y-tauP301L cells treated with a combination of HCQ (10 μM, 24 h), okadaic acid (15 nM, 24 h) or PBA-1105 (0.1 μM, 24 h). Data are presented as mean values ± SD where relevant. P-values (from a two-sided unpaired t test): **P < 0.00821. Source data are provided with this paper.

good albeit rapidly metabolized exposure (Supplementary Table 1), to eradicate misfolded tau aggregates from their brains. The mice were intraperitoneally injected (saline, 20, or 50 mg/kg) three times per week for one month (Fig. 6b). Brain hemispheres were subjected to RIPA-based insoluble/soluble fractionation, followed by immunoblotting analyses of both endogenous wild-type murine tau and human tauP301L. We first confirmed that PBA-1105 AUTOTAC did not degrade endogenous wild-type murine tau in either detergent-soluble or -insoluble fractions (Fig. 6c, d, g, h). In sharp contrast, PBA-1105 induced marked clearance of detergent-insoluble tauP301L in a dosage-dependent manner (Fig. 6c–f). This reduction in insoluble tau aggregates correlated to an increase in RIPA-insoluble LC3 levels (Fig. 6c, d). In contrast to insoluble species, levels of soluble tau species showed little if any reduction (Fig. 6c–f), demonstrating substrate specificity of AUTOTAC towards aggregation-prone tauP301L mutant. Next, we visualized soluble

oligomeric htauP301L-BiFC bodies using Sudan Black B staining on the cross sections of murine brains. Notably, PBA-1105 AUTOTAC induced a drastic reduction of total tau oligomers on both the cortex and the CA1 region of the hippocampus sections in a dosage-dependent manner as determined by both the number of tau bodies and their fluorescence signals (Fig. 6i, k). Moreover, AT8 staining revealed a marked eradication of soluble bodies of phosphorylated tau-BiFC in the cortex as well as the CA1 region (Fig. 6j, l). These results indicate that AUTOTAC provides a platform to eradicate pathological aggregates from the brain.

## Discussion

In this study, we developed AUTOTAC as a generally applicable chemical platform that enables targeted degradation of a variety of cellular proteins. Our previous work has established the ability

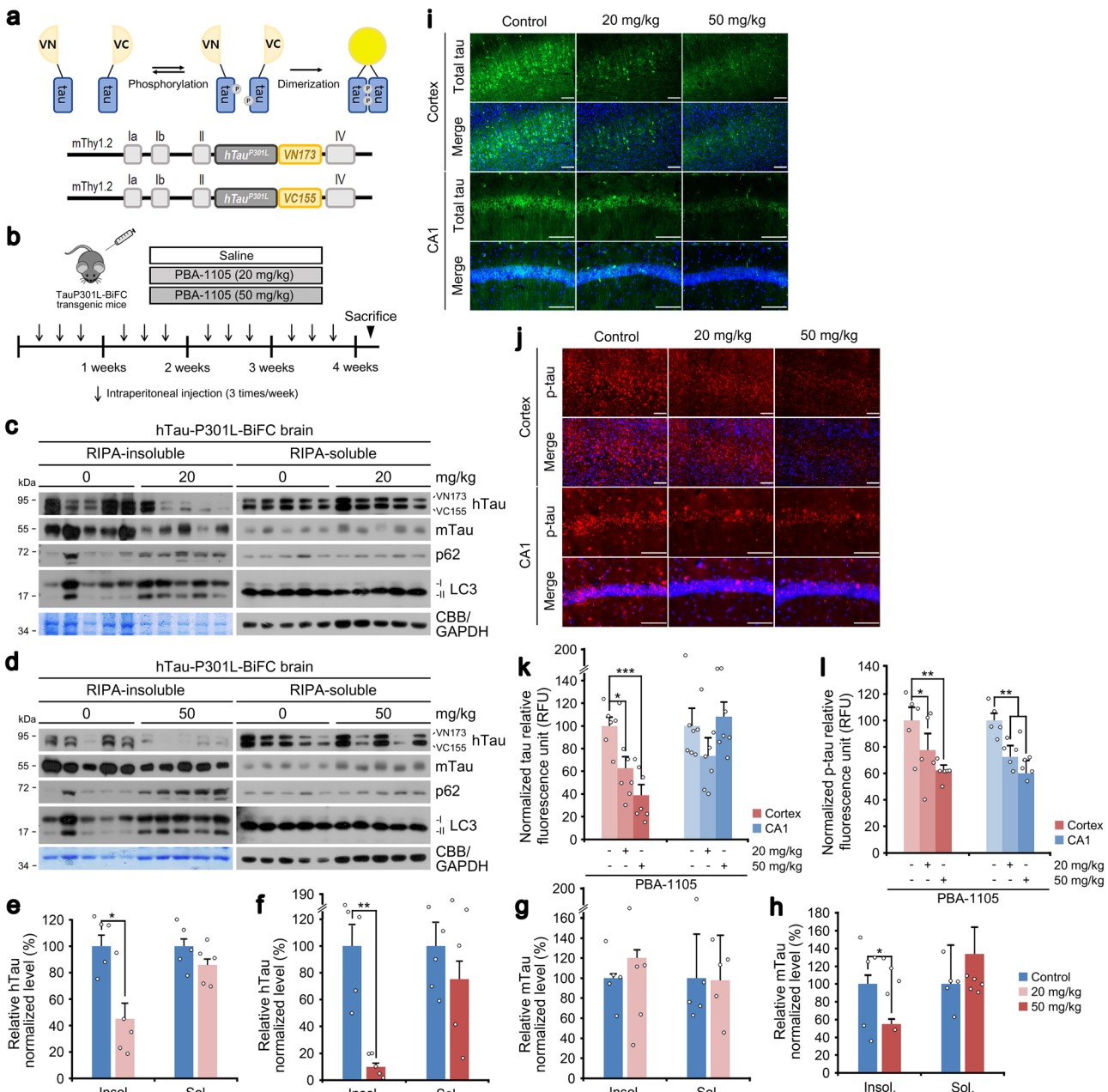

**Fig. 6 Chaperone-based AUTOTAC ameliorates mutant tau pathology in brain-specific murine model. a** Schematic of hTauP301L-BiFC murine model construction. **b** Injection timeline and details of PBA-1105 in hTauP301L-BiFC murine model. **c, d** RIPA-insoluble fractionation assay in brain tissues of hTauP301L-BiFC mice intraperitoneally injected with PBA-1105 (20 or 50 mg/kg). **e, f** Normalized densitometry of (**c**) and (**d**) for hTau levels, respectively (n = 5 mice). **g, h** Same as (**e**) and (**f**) but for mTau levels, respectively (n = 5 mice). **i, j** Immunohistochemistry of BiFC fluorescence for total hTau levels or AT8 fluorescence for total phosphorylated levels in hTauP301L-BiFC mice injected with PBA-1105 as outlined in (**b**). Scale bar, 100 μm. **k, l** Quantification of BiFC or AT8 punctate fluorescence signals in **i** and **j**, respectively (n = 7 mice). Data are presented as mean values ± SEM where relevant. *P*-values (from a two-sided unpaired *t* test): *P < 0.0111 (for insoluble hTau, 20 mg/kg), **P < 0.00105 (for insoluble hTau, 50 mg/kg), *P < 0.0442 (for insoluble mTau, 50 mg/kg). Source data are provided with this paper.

of a p62-binding ATL to activate an otherwise inactive p62 to an autophagy-compatible form via a conformational change. Upon binding to ATL, p62 exposes its PB1 domain for self-polymerization in complex with TBL-bound cargoes (effectively sequestering target cargoes) and its LIR domain for interaction with LC3 on autophagic membranes. Here, we report the proof-of-concept development of AUTOTACs built upon ATL-based p62 binding and activation, through which AUTOTACs can mediate targeted degradation of not only monomeric oncoproteins (whose oncogenic signaling was functionally silenced) but

also oligomeric species of aggregation-prone proteins, including hallmark substrates of neurodegenerative proteinopathies. Therapeutic efficacy of a misfolded protein-targeting AUTOTAC was further confirmed in a brain-specific murine model expressing transgenic human mutant pathological tau. Additionally, AUTOTACs required neither the ubiquitination of the target protein nor the p62-mediated recognition of said ubiquitin chains on the target protein for its sustained degradation. These results substantiate AUTOTACs as generally applicable heterobifunctional chimeric degraders for Ub-independent and p62-mediated

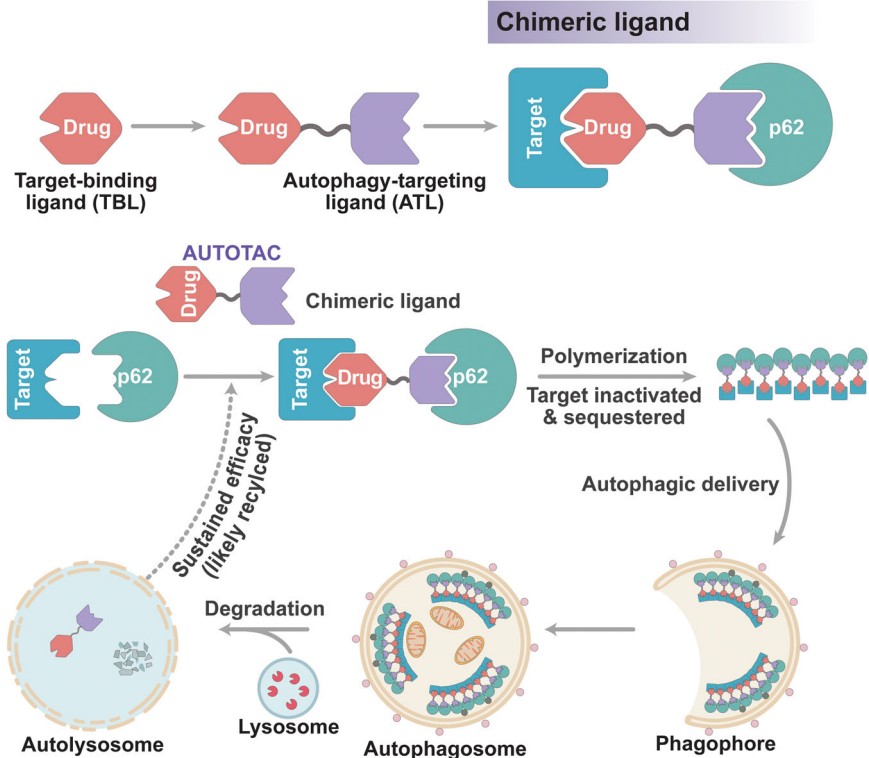

**Fig. 7 Speculative model of AUTOTAC and its mechanism-of-action.** Connecting a protein-of-interest target-binding ligand (TBL) to a p62-binding autophagy-targeting ligand (ATL) using an intermediate linker generates the chimeric AUTOTAC degrader. Recognition of the target protein occurs in tandem with the binding and activation of p62 via its ZZ domain, which is self-polymerized in complex with cargoes. Additionally, such interaction initiates a macroautophagy induction cascade in a p62-dependent manner. AUTOTACs show sustained degradative efficacy post-lysosomal degradation of the target protein, suggesting that it is recycled.

autophagic clearance of a broad range of intracellular target proteins.

Thus, we speculate that AUTOTACs would mediate targeted degradation through the following multi-step mechanisms (Fig. 7). First, AUTOTAC brings a target to p62 via its TBL and ATL, forming a ternary complex. Second, normally inactive p62 is structurally activated for self-oligomerization, forming target-p62 oligomeric complexes. Third, AUTOTAC facilitates Ub-and proteasome-independent degradation of the target-p62 complexes via macroautophagy. Fourth, AUTOTACs are recycled from the lysosome towards other targets in the cytosol, providing a sustained nature of degradation.

Among proteinopathies with gain-of-function toxicity, a large subset is defined as protein misfolding/aggregation disorders wherein misfolded proteins, from their monomeric to oligomeric and aggregated species, are pathological[29,30]. Due to the inherent size and substrate-conformation limitations of a proteasome, any conformationally stable subspecies of a protein beyond its monomeric form is not only non-degradable by the UPS but can even clog the pores of proteasomal subunits[12,31]. While several recent efforts using PROTACs have succeeded in degrading pathological and aggregation-prone tau, it is most likely that the targeted species were monomeric and that the anti-aggregate efficacy, if any, is more of a prevention than a treatment[5]. In this sense, AUTOTACs provide a direct means to target not only the monomeric but also the oligomeric and aggregated species of these pathological hallmark proteins.

Another advantage that AUTOTAC offers over current TPD modalities is in its promiscuity. Notably, the requirement of ligand-induced proximity to achieve spatial and temporal co-localization between E3 and target substrate using PROTACs

necessitates extremely specific linker lengths and types for ternary complex formation, subject to change for each of the numerous E3 ligase-substrate combinations[3,7,8]. Moreover, despite the initial promising outlook on hijacking E3 ligases to ubiquitinate non-native substrates, it is becoming increasingly clear that the current spectrum of E3 ligases used in PROTAC technology exhibits restricted substrate specificities, and that a pan-ubiquitinating E3 ligase is yet to be found[16,40]. Our data imply that AUTOTACs may not be critically reliant upon a specific linker length and do not require ubiquitination of the target substrate for its degradation. These lines of evidence suggest that AUTOTACs may not require protein-protein interaction-mediated positive cooperativity for ternary complex formation, at least for TBLs with high affinity towards their respective substrates. That said, however, some low-affinity TBL-target combinations might benefit from p62-target interaction, including recognition of ubiquitin chains on a substrate. Additionally, the variety of structurally distinct p62-ZZ ligands used in AUTOTACs to successfully eradicate equally numerous and diverse target proteins highlight p62 as a pan-autophagy receptor, capable of targeting non-native or autophagy-resistant substrates such as MetAP2 or hyperphosphorylated mutant tau. Thus, our work opens a line of clinical research exploring alternative avenues of targeting the human proteome for degradation.

Critical remaining questions from our study involve the pharmacological and mechanistic properties of AUTOTACs. For example, it remains to be seen whether and how AUTOTACs can be recycled for multiple rounds of degradation. While AUTOTAC-mediated silencing of a target oncoprotein and its downstream signaling is several folds more effective than that of the target-binding inhibitor alone, and AUTOTACs do indeed

exhibit sustained efficacy, it remains to be seen whether AUTOTACs act catalytically and/or escape the lysosome, not unlike the cytotoxic drug moieties of antibody-drug conjugates[41]. Additionally, the off-target and selectivity issues of the AUTO-TAC platform have yet to be fully investigated and should be addressed in follow-up studies. While our data show selective interaction with p62 over NBR1, which carries a similar ZZ domain, the lipophilic nature of the current generation of ATLs may require further optimization to minimize off-target binding. Another question concerns the autophagic sequestration mode of action possibly unique to AUTOTACs. Given the propensity of p62 for self-oligomerization, it will be interesting to determine whether and how much sequestration of a target protein contributes to the overall efficacy of AUTOTACs in biologically inactivating said target. Although the general efficacy of AUTOTACs has yet to be fully evaluated, our results suggest that activating the p62-ZZ domain in an Arg/N-degron-dependent manner for targeted proteolysis can provide an avenue of research and therapeutic investigation into a myriad of diseases.

## Methods

**Compounds, plasmids, and other reagents**. The chemical synthesis and analytical data of Nt-Arg-mimicking compounds are described in the Supplementary Methods.

The recombinant neurodegenerative hallmark protein plasmid was constructed as follows. The GFP tagged Htt-103 plasmid was constructed into pcDNA3.1/myc-His plasmid (Thermo Fisher Scientific) at EcoRI/XhoI sites using PCR amplification. The TauP301L plasmid was a gift from Min Jae Lee (Seoul National University, Korea). These plasmids were transiently transfected using Lipofectamine 2000 or Lipofectamine 3000 (Thermo Fisher Scientific). Other reagents used in this study were bafilomycin A1, hydroxychloroquine, (Sigma); MG132 (Enzo).

**Transfection**. Plasmids were transfected into HeLa and HEK293T cells using the Lipofectamine 2000 Transfection Reagent according to the manufacturer's instructions (Invitrogen).

**RNA interference analysis**. Cells in a 12-well plate ($0.5 \times 10^6$ per well) were transfected with 40 nM siRNA using RNAiMax reagent (Thermo Fisher Scientific). The sequences of pre-designed Silencer Select siRNAS (Thermo Fisher Scientific) against p62, and of pre-designed siRNA (Bioneer) against siubb or siATG5 are as follows: sip62 (sense, 5′-GUGAACUCCAGUCCCUACA-3′; antisense, 5′-UGUAGGGACUGGAGUUCAC-3′), siubb (sense, 5′-CCAGCAGAGGCUCAU-CUUU-3′; antisense, 5′-AAAGAUGAGCCUCUGCUGG-3′) and siATG5 (sense, 5′- CCUUUCAUUCAGAAGCUGUtt-3′; antisense, 5′-ACAGCUUCUGAAU-GAAAGGtc -3′).

**Antibodies**. The antibodies used in this study are as follows: mouse monoclonal anti-p62 (Abcam, ab56416, 1:10,000), rabbit polyclonal anti-LC3 (Sigma, L7543, 1:10,000), rabbit polyclonal anti-ATE1 (Sigma, HPA038444, 1:1000), mouse monoclonal anti-FK2 specific to Ub-conjugated proteins (Enzo, BML-PW8810, 1:3000), rabbit polyclonal anti-GAPDH (BioWorld, AP0063, 1:20,000), rabbit polyclonal anti-b-actin (BioWorld, AP0060, 1:20,000), mouse monoclonal anti-MetAP2 (Santa Cruz, sc-365637, 1:2000), rabbit polyclonal anti-ERβ (Invitrogen, PA1-310B, 1:2000), rabbit polyclonal anti-Androgen Receptor (Cell Signaling, 3202, 1:5000), rabbit polyclonal anti-EGFR (Cell Signaling, 4265, 1:2000), rabbit polyclonal anti-p-Akt (Cell Signaling, 9271, 1:2000), rabbit polyclonal anti-Akt (Cell Signaling, 2920, 1:1000), rabbit polyclonal anti-p-ERK (Cell Signaling, 9101, 1;1000), rabbit polyclonal anti-ERK (Cell Signaling, 9102, 1:1000), rabbit polyclonal anti-ATG5 (Novus, NB110-53818, 1:1000), mouse monoclonal anti-Ub (Santa Cruz, sc-8017, 1:2000), mouse monoclonal anti-GFP (Santa Cruz, sc-9996, 1:2000), mouse monoclonal anti-Tau5 (Invitrogen, AHB0042, 1:5000), rabbit polyclonal anti-p-Tau (Invitrogen, 44–752G, 1:5000). The following secondary antibodies were used: alexa fluor 488 goat anti-rabbit IgG (Invitrogen, A11034, 1:1000), alexa fluor 488 goat anti-mouse IgG (Invitrogen, A11029, 1:1000), alexa fluor 555 goat anti-rabbit IgG (Invitrogen, A32732, 1:1000), alexa fluor 555 goat anti-mouse IgG (Invitrogen, A32727, 1:1000), anti-rabbit IgG-HRP (Cell Signaling, 7074, 1:10,000), and anti-mouse IgG-HRP (Cell Signaling, 7076, 1:10,000).

**Cells and cell culture**. HeLa, HEK293, HEK293T, U-87 MG, ACHN, MCF7 and LNCaP cell lines were obtained from ATCC. SH-SY5Y-tauP301L-GFP was obtained from Innoprot (P30722). HeLa-NLS/NES-Htt-Q25/Q97-eGFP cell lines were a gift from Min Jae Lee (originally created by Min Jae Lee's lab at Seoul National University, Korea). The above cell lines were cultured in Dulbecco's

Modified Eagle's Medium, MEM medium or RPMI-1640 medium with 10% Fetal Bovine Serum and antibiotics (100 units/mL penicillin and 100 μg/mL streptomycin) in a 5% CO2 incubator. For stable cell lines, the expression of the intended tagged target protein(s) was confirmed by immunoblotting and/or immunocytochemistry.

**Western blotting**. Adherent cells were washed with phosphate-buffered saline (PBS) and cell pellets were lysed in SDS-based sample buffer (277.8 mM Tris-HCl, pH 6.8, 4.4% LDS, 44.4% (v/v) glycerol) containing beta-mercaptoethanol. Alternatively, cell pellets or protein supernatants were lysed in 5X Laemmli sample buffer. Whole-cell lysates were separated using SDS-PAGE, and transferred onto polyvinyllidene difluoride membranes at 100 V for 2 h at 4 °C. The membrane was incubated with a blocking solution consisting 4% skim milk in PBS solution for 30 min at room temperature and incubated with primary antibodies overnight, followed by incubation with host-specific HRP-conjugated secondary antibodies (1:10000 dilution). For signal detection, the membrane was developed with a mixture of ECL solution (Thermo Fisher Scientific) using X-ray films. Densitometry of developed bands was measured and analyzed with ImageJ (NIH, Bethesda).

**Immunocytochemistry**. Cells were cultured on coverslips coated with poly-L-lysine (Sigma) to observe cellular localization of proteins. Using 4% paraformaldehyde in PBS (pH 7.4), the cells were fixed for 15 min at room temperature and washed three times for 5 min with PBS. After fixing, the cells were permeabilized with 0.5% Triton X-100 in PBS solution for 15 min and washed three times with PBS for 5 min. The cells were blocked containing 2% BSA in PBS solution for 1 h at room temperature. Subsequently, the cells were incubated with primary antibody diluted in 2% BSA/PBS solution overnight at 4 °C, followed by washing the cells three times for 10 min with PBS and incubated with Alexa Fluor-conjugated secondary antibody diluted in 2% BSA/PBS for 30 min at room temperature. Using a DAPI-containing mounting medium (Vector Laboratories), the coverslips were mounted on glass slides. Confocal images were taken by laser scanning confocal microscope 510 Meta (Zeiss) and analyzed by Zeiss LSM Image Browser (ver. 4.2.0.121). Subsequently, cells were deemed to exhibit significant co-localization if more than ten clear puncta structures of the respective proteins showed association or full co-localization. Quantification results are shown as mean ± S.D. or S.E.M. of three independent experiments

**In vitro p62 oligomerization**. HEK293T cells were transiently transfected with a plasmid encoding p62-myc/his fusion proteins. Cells were resuspended in lysis buffer [50 mM HEPES (pH 7.4), 0.15 M KCl, 0.1% Nonidet P-40), 10% glycerol, containing a mixture of protease inhibitors and phosphatase inhibitor (Abcam)]. To lyse the cells, 10 cycles of freezing and thawing was done, followed by centrifugation at $13,000 \times g$ for 20 min at 4 °C. Using a BCA assay, the protein concentration of the supernatant was measured. A total of 1 μg of protein was incubated with 1 mM of p62-ZZ ligands in the presence of 100 μM bestatin (Enzo) at room temperature for 2 h. After incubation, each sample was mixed with non-reducing 4X LDS sample buffer, heated at 95 °C for 10 min, and resolved using 4–20% gradient SDS-PAGE (Bio-Rad). To monitor the conversion of p62 monomers into oligomers or aggregates, IB assay was performed using anti-myc antibody.

**In vivo oligomerization**. HEK293T cells were transfected with P301L-tau-EGFP plasmid using Lipofectamine 2000. HEK293T and SHSY5Y-tau cells were treated with p62-ZZ ligands for 24 h. After incubation on ice for 30 min for supernatant collection, the cells were lysed by a cycle of freezing/thawing and centrifuged at 13,000 g for 10 min. Protein concentration was determined using the Pierce BCA Protein Assay Kit (Thermo Fisher Scientific). Non-reducing 4X LDS sample buffer was added to sample lysate, followed by boiling at 100 °C for 10 min and samples were loaded on a 3% stacking and 8% separating SDS-PAGE. Immunoblotting assays were carried out using anti-GFP antibody (Sigma) to visualize the oligomeric complexes of Tau.

**In vitro pulldown assay**. A set of synthetic 12-mer peptides corresponding to the N-terminal sequences of the V-BiP (V[19]-EEDKKEDVGTK-biotin) and V-RGS4 (V[2]-KGLAGLPASCLK-biotin) was C-terminally biotin-conjugated. Alternatively, biotinylated versions of YOK-1304, YT-8-8, YTK-1105, YOK-2204, and PBA were synthesized. To cross-link the above peptides with resin beads, biotin-conjugated peptides, and small molecule were mixed with high-capacity streptavidin agarose resin (Thermo Fisher Scientific) at a ratio of 0.5 mg of peptide per 1 mL of settled resin and incubated on a rotator at 4 °C overnight. After washing five times with PBS, the peptide/small molecule-bead conjugates were diluted with PBS at a 1:1 ratio. To prepare protein extracts, cells were collected by centrifugation and lysed by freezing and thawing at least 10 times in hypotonic buffer [10 mM KCl, 1.5 mM MgCl2, and 10 mM HEPES (pH 7.9)] with a protease inhibitor mix (Sigma). After centrifugation at 15,000 rpm 4 °C for 15 min, proteins were quantified using a BCA protein assay kit (Thermo Fisher Scientific). Total protein (200 μg) diluted in 300 μL of binding buffer [0.05% Tween-20, 10% glycerol, 0.2 M KCl, and 20 mM HEPES (pH 7.9)] were mixed with 50 μL of peptide/small molecule-bead resin and

incubated at 4 °C for 2 h on a rotator. The protein-bound beads were collected by centrifugation at 2400 × g for 3 min and washed five times with binding buffer. The beads were resuspended in 25 µL of SDS sample buffer, heated at 95 °C for 5 min, and subjected to SDS/PAGE and immunoblotting.

**Co-immunoprecipitation**. To study protein interactions, co-immunoprecipitation assays were performed. For exogenous co-IP, HEK293T cells were transfected with indicated constructs using Lipofectamine 2000. For both endogenous and exogenous co-IP, cells were treated after 24 h with specified reagents for indicated incubation times. The cell pellets were scraped and pelleted by centrifugation, were resuspended and lysed in immunoprecipitation buffer [50 mM Tris-HCl pH 7.5, 150 mM NaCl, 0.5% Triton X-100, 1 mM EDTA, 1 mM phenylmethylsulfonyl fluoride (PMSF; Roche) and protease inhibitor cocktail (Sigma)] for 30 min on rotator at 4 °C. Next, the supernatant and remaining pellet were passed through a 26-gauge 1 mL syringe 15 times and centrifuged at 13,000 g at 4 °C and collected for the supernatant, to which we added normal mouse IgG (Santa Cruz) and Protein A/G-Plus agarose beads (Santa Cruz) to preclear the lysate at 4 °C on a rotor overnight. Cell lysate was then incubated with M2 FLAG-affinity Gel agarose beads (Sigma) at 4 °C on a rotor for 3 h. The gel beads were washed four times with IP buffer, resuspended in 2X Laemmli Sample Buffer, separated by SDS-PAGE and analyzed by immunoblotting with specified antibodies.

**Triton X-100-based insoluble/soluble fractionation**. SH-SY5Y-tau BiFC cells were treated with PBA-1105, PBA and YTK-1105 to determine the degraded fraction of Tau. Using a cell lysis buffer (20 mM HEPES pH 7.9, 0.2 M KCl, 1 mM MgCl₂, 1 mM EGTA, 1% Triton X-100, 10% glycerol, protease inhibitor and phosphatase inhibitor), the cells were collected and incubated on ice for 15 min, followed by centrifugation at 13,000 g for 10 min at 4 °C. The supernatant was collected as the soluble fraction and the pellet as the insoluble fraction. Using PBS, the insoluble fraction was washed 4 times and lysed with a SDS-detergent lysis buffer (20 mM HEPES pH 7.9, 0.2 M KCl, 1 mM MgCl₂, 1 mM EGTA, 1% Triton x-100, 1% SDS, 10% glycerol, protease inhibitors and phosphatase inhibitors). 5X Laemmli sample buffer was added to the soluble and insoluble samples and boiled for 10 min at 100 °C and loaded on a SDS-PAGE gel.

**Wound healing assay**. To analyze cell migration in two dimensions, U-87 MG or ACHN cells were plated to a monolayer. Cells were scratched with a sterile 10 µl pipette tip and the debris was removed using medium. Cells were treated with compounds and incubated for different time to monitor cell behavior. Photographs were obtained using a microscope at different time point.

**Cell viability assay**. Cell viability was quantified using the water-soluble tetrazolium salt-based EZ-Cytox cell viability assay kit (Dojindo Laboratory) according to the manufacturer's instructions. Briefly, following siRNA-mediated knockdown of control or ATE1 (48 h), HeLa cells in a 96-well plate were treated with the indicated compounds. Subsequently, assay reagent solution (10 µL) was added to each well and cells were incubated for 4 h at 37 °C in a CO₂ incubator. Optical density (OD) values were measured at 450 nm using a Evolution 350 UV-Vis Spectrophotometer (Thermo Fisher Scientific).

**Flow cytometry**. Cell death and cell cycle arrest were quantified by staining cells with propidium iodide for flow cytometry with a BD FACSCalibur (BD Biosciences) according to the manufacturer's instructions. Briefly, $1 \times 10^6$ cells were incubated with fumagillin-105 (1 µM, 48 h) or negative control DMSO. Cells were collected by centrifugation and fixed in 70% ethanol at 4 °C for 24 h. Cells were washed with PBS and stained with propidium iodide (10 µg/mL) with RNAse treatment at 37 °C for 30 min. DNA content at each cell cycle checkpoint was measured and analyzed using BD CellQuest Pro (BD Biosciences) and ModFit LT Systems (Verity Software House).

**Animals**. hTau-P301L- BiFC and ICR mice were bred and housed in a 12:12 light-dark cycle, pathogen-free, temperature- and humidity-controlled facility with food and water available at Korea Institute of Science and Technology. Animal protocols followed the principles and practices outlined in the approved guidelines and also received ethical approval by the Institutional Animal Care and Use Committee (IACUC) of the Korea Institute of Science and Technology.

**Administration of PBA-1105 toTauP301L-BiFC mice**. To evaluate the effect of PBA-1105 on tau aggregation in vivo, PBA-1105 was intraperitoneally administered to 9-month-old TauP301L-mice (n = 7 per group) with 20 or 50 mg/kg dosage. PBS containing 30% polyethylene glycol (PEG) was used as a vehicle. Twelve total administrations were performed for 4 weeks, three times a week.

**Brain tissue preparation**. Mice were anesthetized by intraperitoneal injection of 2% avertin (2,2,2-Tribromoethanol). Mice were then perfused with 0.9% saline. Brains were rapidly extracted and fixed with PBS containing 4% paraformaldehyde for 48 hr. For, cryoprotection, the brains were infiltrated with PBS containing 30% sucrose at 4 °C until they sunk. For cryosectioning, the brains were embedded with O.C.T (Leica, Germany) and serially cut in the coronal plane into 30-µm thick sections on a cryostat microtome (Leica). Tissue slices were transferred to PBS containing 0.05% sodium azide as a preservative and stored at 4 °C.

**Sudan Black B stain and BiFC image acquisition**. To reduce autofluorescence of brain tissues, Sudan Black B stain was performed. Brain tissue slices were mounted onto glass slides and were stained with Sudan Black B solution (70 % ethanol containing 0.05% Sudan Black B) for 10 min. Then, to eliminate the excessive stain, the slides were dipped in PBS containing 0.1% Triton X-100 three times and washed with distilled water after. For nuclei counter-stain, brain tissues were stained with 0.5 µg/mL Hoechst in distilled water for 30 min. BiFC fluorescence ($\lambda_{ex} = 460-490$ nm and $\lambda_{em} = 500-550$ nm) of brain slices were imaged using ZEISS® Axio Scan.Z1 (Zeiss, Germany). Fluorescence and total area of each puncta were measured using ImageJ (NIH, Bethesda).

**Statistics and reproducibility**. For all data shown, stated values represent the mean ± S.D or S.E.M. of at least three independent experiments (unless otherwise stated). For each experiment, sample size (n) was determined as stated in the figure legends. For all experiments, p-values were determined using two-tailed, unpaired student's $t$ test (degree of freedom = $n - 1$) with Prism 6 software (GraphPad).

**Reporting summary**. Further information on research design is available in the Nature Research Reporting Summary linked to this article.

## Data availability
Data generated in this study are provided in the article and its associated files. Source Data are provided with this paper. All other data are available from the corresponding authors on request. Source data are provided with this paper.

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

## Acknowledgements
The authors thank the Y.T.K. and Y.K.K. laboratories' members and AUTOTAC Bio Inc.'s employees for their comments during this study. This work was supported by the National Research Foundation of Korea (NRF) grant funded by the Korea government (MSIT) (NRF-2020R1A5A1019023 and NRF-2021R1A2B5B03002614 to Y.T.K.), the Korea Health Technology R&D Project through the Korea Health Industry Development Institute and Korea Dementia Research Center (KDRC) funded by the Ministry of Science, ICT, and Future Planning (MSIP) (HU21C0201 to C.H.J.), the NRF grant of Korea government (MSIT) (NRF-2021R1A2C3004965), the R&D Convergence Program of NST (CAP-16-03-KRIBB) and by the Korea Research Institute of Bioscience and Biotechnology Research Initiative Program (to B.Y.K.), and the National Research Foundation of Korea (NRF) grant funded by the Korea government (MSIT) (2021R1A2C2093734) and KIST Institutional Program (Atmospheric Environment Research Program, 2E31700).

## Author contributions
Immunoblotting and immunocytochemical assays and their analyses in cell lines were performed by C.H.J., H.Y.K., M.J.L., D.Y.P., C.A.J., E.B.Y., and S.H.P.; ATLs binding the p62-ZZ domain were synthesized and modeled by S.G., K.Y.K., J.E.N., and H.T.K.; murine injections, brain tissue preparation, and immunohistochemistry were performed by W.S.Y., S.L., and S.G.S.; mice sacrifice and tissue collection for western blotting were carried out by C.H.J., M.J.L., and C.A.J.; in vitro pulldown assays were carried out by A.J.H., S.B.K., and M.J.L.; wound healing assays were done by C.H.J. and H.Y.K.; in vitro p62 and in vivo tau oligomerization assays were carried out by C.H.J., H.Y.K., M.J.L., C.A.J., and E.B.Y.; Y.T.K., B.Y.K., Y.K.K., J.I.K., and H.M.C. provided guidance, specialized reagents, and expertise. C.H.J. and Y.T.K. designed the experiments and wrote the paper.

## Competing interests
Seoul National University and AUTOTAC Bio, Inc. have filed patent applications (C.H.J., H.Y.K., M.J.L., A.J.H., S.G., J.E.N., H.T.K., and Y.T.K.; US 17/262,157 undergoing continuation-in-part, PCT/KR2019/009205 under examination; proof-of-concept AUTO-TAC platform) based on the results of this study. The remaining authors declare no competing interests.
