## [Peer Review File · Nature Communications]

REVIEWER COMMENTS

Reviewer #1 (Remarks to the Author):

In this contribution, Ji et al. reported a novel autophagy-targeting chimera (AUTOTAC) system. This work is a natural extension of the previous pioneering work on p62 and autophagy by the groups of investigators. Protein degraders are emerging therapeutic modalities with PROTACs as the most advanced platform. However, the efficacy of most PROTACs heavily rely on the formation of ternary complexes, rendering the rational design of PROTACs highly challenging. Additionally, protein aggregates, which are usually cleared through the autophagy-lysosome pathway, cannot be degraded by PROTACs. This work has the potential to address the two aforementioned major downsides of PROTACs, which is considered highly significant. Unfortunately, this manuscript is poorly written with many mistakes and mislabeled figures, making it very difficult to understand in some cases. I'd like to re-review this manuscript after a major revision.

Some major questions:

1. Please include the chemical structures of all the compounds used in this study either in main figures or SI. For example, Figure 1c has YTK-1105. But only YTK-105 structure is shown in Figure 1b. Is 1105 a new compound or typo?
2. Why different ATLs are used in ER- β , AR, and MetAP2 AutoTACs and Fig.4b (misfolded protein AutoTACs)? If these are optimized AutoTACs, please include the comparison of different ATLs in SI.
3. PROTACs usually require extensive optimization of linker length and the exit vector direction of the linker. As the authors pointed out, AUTOTACs are more flexible in molecular design. To demonstrate this point, one AUTOTAC should be picked as an example with different linker lengths and different exit vector to compare their degradation efficacy.
4. In Figure 3a and 3c, both AR and LC3 are diffusive without AutoTAC treatment. How was the colocalization quantified? Based on counting punctates?
5. In Figure 3f, why did p62 and ATG5 siRNA knockdown significantly increase ER- β levels? Does this suggest that endogenous ER- β degrades through the autophagy pathway?
6. In Figure 3m, why were ACHN cells used for ER- β degraders?
7. In Figure 3n and other prostate cancer studies, enzalutamide (anti-androgen standard of care) and ARV-110 (AR PROTAC degrader) should be used for comparison. ARV-110 has IC50 in low nM range in VCaP cells based a proliferation assay. Is the relatively low potency of YOK-2204 due to incomplete degradation of AR?
8. For all the compounds, please include Proton and Carbon NMR spectra for all the new compounds in the SI. Please include high resolution Mass spec data for all new compounds.

9. What is the PK profile for PBA-1105? How the dosing regimen (3 times per week) was determined?
10. In Figure 6, tau aggregates are significantly reduced at 50 mpk. What are the cognitive changes of these treated mice?
11. Many figure legends lack sufficient details to understand the experiments. Please add error bars and statistical analysis if missing.
12. Line 151, it should be "YTK-105"?

Reviewer #2 (Remarks to the Author):

This manuscript by Ji et al. describes the design of degraders containing a p62-binding ligand and the application to the degradation of aggregate-prone proteins, such as Tau.

AUTAC (Takahashi et al., Mol Cell, 2019) and ATTEC (Nature, 2019) have been reported as autophagy-based degraders. AUTAC has successfully degraded cytoplasmic proteins and dysfunctional mitochondria. ATTEC is a molecular glue-type degrader that successfully degraded mutantHTT aggregates in a mouse model.

Ji et al's degrader AUTOTAC is the second example of a degrader that degrades protein aggregates. Similar to AUTAC, it is a hetero-bifunctional degrader, which allows to design for a variety of substrates by the selection of a target-binding ligand (TBL). Their degradation tag binds to p62/sqstm1 and generates insoluble aggregates of p62. (I think this tag should not be called autophagy-targeting ligand: ATL, because p62 is not autophagy-specific; p62-ligand is more appropriate). In this case, the substrate bound to the target-binding ligand (TBL) crashes out with p62. This insoluble material is degraded by autophagy.

(It is not appropriate to describe this insoluble p62 aggregate as an autophagy-compatible form; Abstract, line 128, 141)

Although the AUTOTAC approach is unique and has potential, however, the manuscript requires significant strengthening to support the conclusion that AUTOTAC degrades cargos by "activating p62 and enhancing autophagic flux".

Based on the UPS, so many degraders have been reported. Some of these also inhibit protein aggregates associated with neurodegenerative diseases such as mHTT (eg. S. Tomoshige et al., Angew. Chem. Int. Ed. 2017, 56, 11530). Perhaps, though, by degrading them at the monomer or oligomer stage. For this manuscript to satisfy Nature Commun readers, authors would need to

provide clear evidence that AUTOTAC is different from the UPS-based degraders. Unfortunately, the current data do not sufficiently support the conclusion. I would like to discuss a few important issues here.

1) The authors argued the co-localization of LC3 with p62 as evidence for isolation membrane recruitment, but this is not convincing. p62 has a LIR motif that binds to LC3, so puncta that co-localize LC3 and p62 are not necessarily autophagosomes. It could be a membrane-free p62 aggregate or droplet. The WB experiments with DHQ are not fully convincing as an evidence for the autophagic mechanisms. The authors should examine if the lysosomal acidification of the AUTOTAC's target proteins. For example, it needs to be shown that hTau aggregates, which fused with pH-sensitive fluorescence, are acidified as a result of an AUTOTAC treatment.

The presence of an isolation membrane around the aggregates can be most reliably verified using electron microscopy. Alternatively, it may be useful to analyze the localization of proteins that act during autophagosome formation, such as WIPI2.

5) The authors do not distinguish the following mechanisms. "Aggregation-binding AUTOTAC" may recruit p62 to intracellular aggregates (e.g. Tau) for degradation as proposed in this manuscript. But it is also needed to consider that "aggregation-binding AUTOTAC" may newly form aggregates from p62 and soluble monomeric or oligomeric Tau. If the former mechanism exists, bilayered aggregates of Tau and p62 may be observed. The latter mechanism would also exist, if this is the case, colocalization data of LC3 with Tau-GFP alone (Fig. 5I) would not sufficiently support the model that AUTOTAC degrades protein aggregates. The reduction of Tau aggregates by AUTOTAC is nevertheless important, however, UPS-based degraders are also able to reduce protein aggregates by removal of aggregate-prone proteins at monomeric or oligomeric states.

6) In Figure 4, the authors used the proteasome inhibitor MG132 to accumulate ubiquitinated proteins and examined the effect of AUTOTAC on the accumulated ubiquitin signal density. The title of Figure 4 includes "targeted delivery and degradation of misfolded protein cargos". It is quite misleading to describe as if all of the ubiquitinated proteins accumulated by MG132 treatment are misfolded. Ubiquitination is a posttranslational modification also involved also in signal transductions. The data in Fig. 4 in general do not support sufficiently the hypothesis that AUTOTAC delivered misfolded proteins to autophagy, and therefore requires revision.

Specific points

1) line 62: What does "autophagy-compatible form" means? There are no experimental data. If it means p62 aggregate, simply state it that way.

2) line 69: "AUOTAC"

3) Line 99: The cited reference 12 does not related to “neurodegenerative proteinopathies”. Please cite more appropriate reference here.

4) This sentence is NOT sufficient to describe the recent progresses of autophagy-based degraders. The reference for AUTAC is cited but it does not explain what has been done such as removal of impaired mitochondria. Most importantly, the authors did not cite nor mention the ATTEC technology here (Nature, 2019).

Moreover, a phrase of “targeting extracellular and secreted/membrane proteins directly to the lysosome (line 108) “is related to the LYAC technology of Dr. Bertozzi but their work is not cited here (Banik, S. et al. Lysosome-targeting chimaeras for degradation of extracellular proteins. Nature <https://doi.org/10.1038/s41586-020-2545-9> (2020)).

5) line 124: What are the criteria for “generally applicable chemical tool”? The criterion should be clearly indicated prior to the discussion if AUTOTAC meets this criterion or not. How about the previous techniques such as ATTEC?

6) line130: This reviewer understands that the ATL alone induces aggregation of p62. However, I do not understand why you suddenly asserted that AUTOTAC can induce the degradation of a broad range of substrates. I do not agree with the authors' notion that “ATL-mediated autophagic activation of p62 does not need to form a ternary complex”. In fact, the data in the following Results section seemed to suggest that AUTOTAC molecule binds with two proteins (substrate and p62) to form a ternary complex.

7) line 141: I do not understand clearly what is “autophagy compatible form”. It is “insoluble aggregate of p62”?

8) line 143: “YTK-F105, exhibited efficacy and selectivity to activate p62”

I could not find any data regarding the efficacy of these four compounds neither in Fig. 1b nor Suppl Fig. 1. The data in Fig. 1 showed that these compounds bind to p62 and induce p62 aggregation. But it does not mean that these compounds are selective to p62.

I also do not understand what the authors mean by p62 activation. The data shows that the compounds induced aggregation of p62 and there is no other evidence that the aggregated p62 are “activated”.

9) line 150: This pulldown assay showed YOK-1302 and YTK-1105 bind to p62. But specificity of these compounds to p62 are not shown in this manuscript. Considering the highly lipophilic structure of ATLs, this reviewer has a concern on their specificities to p62.

10) line 153-157: The colocalization of p62 and LC3 proteins should not be used as the sole evidence of autophagy. Because p62 has an LIR domain, it is known that p62 and LC3 often colocalize in punctate (aggregates or condensates) without an autophagic membrane (see Komatsu, Cell 2007, 131, 1149 & Pankiv, JBC 2007, 282, 24131). Therefore, the notions such as “(ATLs) target p62 to autophagosome membrane” or “(ATLs) facilitate autophagosome biogenesis# are not sufficiently supported by the data presented.

To show that ATL induces autophagy (biogenesis of isolation membrane), experiments with cell lines that lack an autophagy-initiation complex is required. Deletion of Atg13 or FIP200 are frequently used to examine if a treatment induces autophagy. Mizushima reported a quantitative method to analyze the autophagy levels in culture cells and living animals (Kaizuka et al., Mol Cell, 2016, 64(4):835-849).

The existence of membrane of autophagosome at the ATL-induced p62 aggregates may be examined with the colocalization of Syntaxin-17 or electron microscopy.

11) 9) line 157-160: The authors rely heavily on an WB-based autophagic flux assay. This is not fully convincing. Please see the discussion on the major points above.

12) line 162-170: "We speculated....First, AUTOTAC brings a target to p62 via....Secondary,..."

Some of these speculations are not examined experimentally in this manuscript (e.g. catalytic action of AUTOTAC). Please focus on the topics you are going to examine in the following result section.

13) line 171: "These mechanisms are independent of PPI.... thus are generally applicable for a broad range of intracellular protein"

It is too assertive. Please note not all of the speculated mechanisms are fully examined in the manuscript.

14) line 179: "cells at (Fig. 2f-i)"

Please remove "at".

There are multiple errors in panels of Fig. 2 and I was not able to examine fully if the data support conclusion of the authors. Some of the errors include following 15)-19).

15) line 178: "(Fig. 2d,e)"

I guess this should be corrected as "Fig. 2f,g"

16) line 179: "(Fig. 2f-i)"

I guess this should be corrected as "Fig. 2j,i".

17) line 187: "Fig. 2j-l"

I guess this should be corrected as "Fig. 2k,l"

18) line 189: "Fig. 2m,n"

I guess this should be "Fig. 2d,e".

19) line 190: "Fig. 2i,l,o"

This may be corrected as "Fig. 2n, m,o".

20) lines 190-191: "These results validate AUTOTAC as a general chemical tool for targeted proteolysis."

I was so confused by the errors in Figure 2 but it seems like levels of three protein targets decreased in WB analyses. Is this the sole reason by which the authors concluded this is a "general" chemical tools?

21) line 193: vinpocetine-2204 induced the formation of AR+p62+ complexes and AR+LC3+ autophagic membranes (Fig. 3a,b).

Fig. 3a and b showed colocalization of AR and LC3 but the data for colocalization with p62 is not presented.

Moreover, the colocalization with LC3 signals does not always means the existence of "autophagic membrane". LC3 colocalizes with p62 aggregates or droplets without "autophagic membrane". To support the authors conclusion, existence of autophagic membrane must be examined by additional experiments. Thus, the conclusion is not sufficiently supported.

22) line 193-196: PHTPP-1304.....(Fig. 3c,d).

Dose-dependent formation of p62/ERB puncta was shown but this dots formation and colocalization ratio of LC3 (Fig. 3d) is not a convincing indicator of autophagy flux.

To demonstrate the lysosomal degradation of the cargos, use of pH-sensitive protein probes such as GFP-RFP, Keima, or Rosella provides convincing results.

To demonstrate the autophagy induction (increase of autophagy flux) during the treatment with AUTOTAC compounds, the use of Atg13 or FIP200 knockout cells (NOT knockdown) is recommended.

23) line 197 Fig. 3: MetAP2 level in the Fuma.-105 (+) & Baf A1 (+) lane is significantly higher than that in the Fuma.-105 (-) & Baf A1 (+) lane. Why?

24) line 199 Fig. 3f: Please add imaging data of the cells under p62 or Atg5 KO conditions to examine the numbers and characters of punctates.

25) line 299 Fig. 3k, l: These data are obtained with Fuma-105, however, the figure legends describe as "treated with PHTPP-1304, PHTPP".

26) line 247: How the molecules "activated" p62? What does "activation" means?

27) line 248 (important): There are no evidence shown in the Fig. 4c regarding the coexistence of cargoes with oligomeric p62.

28) line 250 Fig. 4d: The level of ubiquitinated proteins decreased in PBA1105(+) & HCQ(+) lane. Do you mean the decrease is not mediated by lysosomal degradation?

- 29) line 253: Please add the number of p62-dots before and after the AUTOTAC treatments. This should be an important factor whenever you discuss the colocalization ratio of something with p62 dots.
- 30) line 254: Existence of autophagic membrane is not shown with experimental evidence.
- 31) line 256: "UPS-resistant misfolded protein aggregates" are not appropriately analyzed in Fig.4. For example, there are no data regarding the numbers of aggregates. Moreover, the authors need to consider aggregates may be generated by the action of AUTOTAC via p62 destabilization.
- 32) line 263: The level of ubiquitinated proteins decreased in MG132(+), Anie-F105(+) & HCQ(+) lane. Do you mean the decrease is not mediated either by lysosomal degradation or proteasomal degradation?
- 33) line 264 Fig 4k: The increase of FK2 colocalization with p62 alone does not sufficiently support the conclusion that the compounds work via "p62-dependent macroautophagy".
- 34) line 277 Supplemental Fig. 5c,d: The data do not exclude a possibility that 4-PBA may promiscuously bind to proteins.
- 35) line 288: This colocalization analysis alone is not sufficient to support their conclusion that PBA-1105 AUTOTAC selectively induced the sequestration and autophagic targeting of tauP301L.
- 36) line 308: To prove that the effect of AUTOTAC on aggregate-prone protein is independent of polyubiquitin, it is better to use E1 inhibitor (e.g. pyr41).
- 37) Lysosomal acidification of Tau aggregates must be monitored with pH-sensitive fluorescent probes.
- 38) line 313: Macroautophagy works only at cytoplasm. How could AUTOTACs degrade Htt-NLS-GFP?
- 39) line 325 Supplemental Fig. 6k,l: Colocalization of cargo with LC3 is not the conclusive evidence that AUTOTAC selectively promoted autophagic targeting. Please add the data with pH-sensitive fluorescent probes (eg. mHtt fusion protein with mCherry-GFP or Rosella) and demonstrate that AUTOTAC surely accelerates the lysosomal acidification of mHtt dots.
- 40) line 339: The selective reduction of hTau over murine WT Tau is remarkable. Please add the data showing the compound selectively bind hTau over murine WT Tau.
- line 408: "It is highly likely..." It is too speculative.

Reviewer #3 (Remarks to the Author):

In this manuscript, Ji et al. developed a new technology, termed AUTOPhagy-Targeting Chimera (AUTOTAC), as a chemical tool that can degrade specific proteins by autophagy. AUTOTAC is an artificial molecule composed of two parts: (1) target-binding ligand that interacts with specific substrates, and (2) autophagy-targeting ligand that binds with the ZZ domain of p62, a selective substrate of autophagy. The authors developed multiple AUTOTACs targeting several oncoproteins and degradation-resistant aggregates and showed that these AUTOTACs could induce degradation of these targets in culture cells and in the brain. This study provides a potentially useful tool that can compensate the previously established other targeted protein degradation technologies (e.g., PROTAC and AUTAC). Overall, the experiments are well performed, but there are some concerns about the effects of AUTOTACs on autophagy and p62.

Major concerns

1. In this manuscript, the effect of autophagy-targeting ligands on autophagy flux is not clearly demonstrated. In Fig. 1h-k, the authors showed that YOK1304 and YTK105 increased the autophagy flux using a lysosomal inhibitor hydroxychloroquine (HCQ). However, the results are not convincing because the amount of p62 and LC3-II under HCQ treatment increased more in YOK1304- and YTK105-treated cells than in control cells. Such changes could result from enhanced transcription, translation and/or protein stability (e.g., by suppression of proteasomal degradation) of p62 upon treatment with YOK1304 and YTK105. To convincingly support their hypothesis that YOK1304 and YTK105 enhance the degradation of p62 by autophagy but not proteasomal degradation, the authors should perform experiments using autophagy-deficient cells and proteasomal inhibitors as the authors have previously performed similar experiments for other ligands (Cha-Molstad et al., *Nat Cell Biol.* 2015). In these experiments (including autophagy flux assays in Fig. 1h-k), the authors should include YT-8-8 and YOK-2204, latter of which is used in later parts of this study but not included in Fig. 1h-k, and quantitatively and statistically analyze the results.

2. Related to above concern, in Fig. 3f, the authors showed that knockdown of p62 and ATG5 abolished the PHTPP-1304-induced degradation of ER β . However, the results are confusing because the amount of Er β strongly increased in p62- and ATG5-depleted cells even without PHTPP-1304 treatment. The authors should quantitatively and statistically analyze the results and discuss the possible causes of this phenomenon. The authors should also perform similar experiments using p62- and ATG5-depleted cells in other ligand-substrate pairs (e.g., Fuma.-105-MetAP2, Vnc-2204-AR, and PBA-1105-Tau) to investigate whether these substrates are indeed degraded dependently on p62 and autophagy.

3. The authors hypothesized that autophagy-targeting ligands interact with the ZZ domain of p62. It can be experimentally tested using p62 mutant lacking the ZZ domain in Fig. 1c.

Minor comments:

There are many mistakes in the text, figures, and figure legends:

P6, line 151, YOK-1304 is not shown in Fig. 1c.

P6, line 151, YTK-1105 is not shown in Fig. 1b.

P7 line 178, "Fig. 2d, e" should be "Fig. 2f, g".

P7 line 179, "Fig. 2f-i" should be "Fig. 2i, j".

P7 line 180, "Fig. 2d, e" should be "Fig. 2f, g".

P7, line 181, "Fig. 2h, i" should be "Fig. 2n, p".

P7, line 187, "Fig. 2j-l" should be "Fig. 2k, l".

P7, line 189, "Fig. 2m, n" should be "Fig. 2d, e".

P7, line 189, "Fig. 2n" should be "Fig. 2h".

P7, line 190, "Fig. 2i, l, o" should be "Fig. 2e, m, p".

P7, line 193, there is no data corresponding to "the formation of AR+p62+ complexes".

P8, line 202, "Supplementary Fig. 4a" should be .

P9, line 220, Fig. 3J is not cited.

P10, line 253, "Fig. 4h, k" should be "Fig. 4h, i".

P11, line 285, "Fig. 5j" should be "Fig. 5i, j".

P12, line 298, "6 and 8" should be "7 and 8".

P12, line 315, "Supplementary Fig. 6c-d" should be "Supplementary Fig. 6a, b".

P12, line 317, "Supplementary Fig. 6e" should be "Supplementary Fig. 6c".

P12, line 318, "Supplementary Fig. 6a, b" should be "Supplementary Fig. 6d, e".

P13, line 320, "Supplementary Fig. 6b, d" should be "Supplementary Fig. 6b, e".

P43, line 975, "YOK-1104" should be "YOK-1106".

P52, line 1040, "YOK-1104" should be "YOK-1106".

P53, line 1054, "YOK-1104" should be "YOK-1106".

Supplementary Fig. 4b is not cited in any parts of this manuscript.

REVIEWER 1

Remarks to the Author: In this contribution, Ji et al. reported a novel autophagy-targeting chimera (AUTOTAC) system. This work is a natural extension of the previous pioneering work on p62 and autophagy by the groups of investigators. Protein degraders are emerging therapeutic modalities with PROTACs as the most advanced platform. However, the efficacy of most PROTACs heavily rely on the formation of ternary complexes, rendering the rational design of PROTACs highly challenging. Additionally, protein aggregates, which are usually cleared through the autophagy-lysosome pathway, cannot be degraded by PROTACs. This work has the potential to address the two aforementioned major downsides of PROTACs, which is considered highly significant. Unfortunately, this manuscript is poorly written with many mistakes and mislabeled figures, making it very difficult to understand in some cases. I'd like to re-review this manuscript after a major revision.

Note: In response to your valuable comments, we performed a series of additional experiments, the results of which are now presented in the revised manuscript as **Figures 1b, 2b, 4b, Supplementary Figures 4g, 4h, 6c, 8** and **Materials & Methods: H-NMR, C-NMR & LC-MS** along with appropriate changes to the text.

Major comments

Comment 1. Please include the chemical structures of all the compounds used in this study either in main figures or SI. For example, Figure 1c has YTK-1105. But only YTK-105 structure is shown in Figure 1b. Is 1105 a new compound or typo?

Reply: The mislabeled text has been corrected (YTK-1105 to YTK-105 and YOK-1104 to YOK-2204; Figures 1b and 1c). The chemical structures of all the compounds are now shown as **Figs. 1b** (for ATLS), **2b** (for anti-oncoprotein AUTOTACs) and **4b** (for anti-aggregate AUTOTACs).

Comment 2. Why different ATLS are used in ER- β , AR, and MetAP2 AutoTACs and Fig.4b (misfolded protein AutoTACs)? If these are optimized AutoTACs, please include the comparison of different ATLS in SI.

Reply: Different ATLS were used to create chimeric AUTOTAC degraders against different cellular target proteins to not only show the diversity of the AUTOTAC platform in targeting p62, but also take into consideration the differing efficacies of the ATLS. We agree with the reviewer that the different ATLS should be compared, and thus have shown the comparison of these different ATLS in the original manuscript by their interaction to p62 (Fig. 1c), efficacy of inducing p62 self-oligomerization (Fig. 1d), p62⁺LC3⁺ puncta formation/co-localization (Fig. 1e, f, g), and induction of cellular autophagy flux (Fig. 1h, i) in the original manuscript. The AUTOTACs presented in this paper are not yet optimized and represent a proof-of-concept batch of degraders.

Comment 3. PROTACs usually require extensive optimization of linker length and the exit vector direction of the linker. As the authors pointed out, AUTOTACs are more flexible in molecular design. To demonstrate this point, one AUTOTAC should be picked as an example with different linker lengths and different exit vector to compare their degradation efficacy.

Reply: In response to this comment, we newly synthesized a PBA-based AUTOTAC compound (PBA-1105b) that contains a linker composed of 7 PEG moieties (**Fig. 4b; page 13, lines 330-334**), in contrast to PBA1105 with 3 PEG moieties. We also confirmed that PBA-1105b with a longer linker induced the degradation of mutant hTauP301L at a similar efficiency as the original PBA-1105 (**Supplementary Fig. 6c; page 13, lines 330-334**). This result provides a strong evidence that in contrast to PROTAC, AUTOTAC does not need a tertiary complex between p62 and a target protein.

Comment 4. In Figure 3a and 3c, both AR and LC3 are diffusive without AutoTAC treatment. How was the colocalization quantified? Based on counting punctates?

Reply: The co-localization was quantified based on counting number of cells that contained a high number (more than 10) of AR⁺LC3⁺ or ER β ⁺LC3⁺ punctate structures. As the reviewer pointed out, we noticed that while LC3, AR and ER β were diffusive without AUTOTAC treatment, they formed punctate structures in a dosage-dependent manner upon AUTOTAC treatment.

Comment 5. In Figure 3f, why did p62 and ATG5 siRNA knockdown significantly increase ER- β levels? Does this suggest that endogenous ER- β degrades through the autophagy pathway?

Reply: We interpret that ER β levels were significantly up-regulated upon siRNA-mediated interference of p62 and ATG5 because endogenous ER β is indeed degraded via autophagy. This is in agreement with other findings that show the autophagic degradation of nuclear receptors, including ER α and androgen receptor (15' *Cell Signal.* 27: 1994; 14' *PLOS One.* e94880; 11' *J Biol. Chem.* 286: 22441). Given that proteasomal degradation of nuclear receptors is also reported, we believe that both the UPS and autophagy mediate degradation of nuclear receptors, including ER β . To further address this comment, we performed additional experiments and now show that up-regulated levels of ER β following proteasomal inhibition are reduced by AUTOTAC treatment (**Supplementary Fig. 4g; page 9, lines 225-229**). Notably, the degradative efficacy persisted for up to at least 8 hours post-washout, indicating that the AUTOTACs are recycled from the lysosome and are thus catalytic in nature (**Supplementary Fig. 4h; page 9, lines 229-231**).

Comment 6. In Figure 3m, why were ACHN cells used for ER- β degraders?

Reply: While ER α is known as an oncoprotein in breast cancer, ER β has a more controversial role in breast cancer (it has been reported as an oncoprotein and a tumor suppressor protein by different groups). However, to date, for renal cell carcinoma, prevailing evidence seems to suggest that ER β is an oncoprotein. Thus, we decided to test the degradative efficacy of ER β -degraders in renal cell carcinoma cell lines (e.g., ACHN cells).

Comment 7. In Figure 3n and other prostate cancer studies, enzalutamide (anti-androgen standard of care) and ARV-110 (AR PROTAC degrader) should be used for comparison. ARV-110 has IC₅₀ in low nM range in VCaP cells based a proliferation assay. Is the relatively low potency of YOK-2204 due to incomplete degradation of AR?

Reply: We hypothesize that the low cell proliferation-inhibiting potency of vinclozolin-2204 AUTOTAC (IC₅₀: 4.7 μ M) compared to ARV-110 PROTAC is indeed due to incomplete and less effective degradation of endogenous AR (corroborated by the approximately 200-fold higher DC₅₀ value

of vinclozolin-2204 compared to ARV-110).

Comment 8. For all the compounds, please include Proton and Carbon NMR spectra for all the new compounds in the SI. Please include high resolution Mass spec data for all new compounds.

Reply: As pointed out, we have now included proton, carbon NMR spectra and high-resolution mass spectrometry data for all novel compounds in **Materials & Methods**.

Comment 9. What is the PK profile for PBA-1105? How the dosing regimen (3 times per week) was determined?

Reply: We analyzed the plasma concentrations and PK parameters of PBA-1105 in male ICR mice in three routes: 20 mpk P.O., 10 mpk I.P., and 5 mpk I.V. The results (page 15, lines 20-22) are now presented as **Supplementary Figure 8 (page 15, lines 391-393)**. The dosing regimen was determined based on both the known literature for injecting 4-PBA in mice and internal data and know-how on the *in vivo* efficacy of ATLS.

Comment 10. In Figure 6, tau aggregates are significantly reduced at 50 mpk. What are the cognitive changes of these treated mice?

Reply: As described in the original manuscript, PBA-1105 was shown to induce the degradation of tau aggregates in mouse brains at 20 and 50 mpk. Due to technical and experimental constraints related with COVID-19 crisis, we were not able to properly assess the cognitive changes of mice treated with PBA-based AUTOTACs. Nonetheless, in a separate project, we developed an AUTOTAC compound (ATC-102) that induces degradation of tau in murine brains at 20 mpk P.O (total 8 administrations) and improves behavior in terms of migratory activity at 10 mpk P.O. (total 18 administrations), which can be presented to you upon request. We feel that the results from ATC-102 is beyond the scope of this study and, thus, will be published at the later time.

Comment 11. Many figure legends lack sufficient details to understand the experiments. Please add error bars and statistical analysis if missing.

Reply: As kindly pointed out, we have made changes throughout the figure legends to better explain the experiments (including matching each legend to its figure). Error bars and statistical analyses, where missing, have been included, and are described in either the legend itself or as a collective in the Statistical analysis section of Materials and Methods.

Comment 12. Line 151, it should be “YTK-105”?

Reply: The typo is now corrected (**page 6, line 155**).

REVIEWER 2

Remarks to the Author: This manuscript by Ji et al. describes the design of degraders containing a p62-binding ligand and the application to the degradation of aggregate-prone proteins, such as Tau. AUTAC (Takahashi et al., Mol Cell, 2019) and ATTEC (Nature, 2019) have been reported as autophagy-based degraders. AUTAC has successfully degraded cytoplasmic proteins and dysfunctional mitochondria. ATTEC is a molecular glue-type degrader that successfully degraded mutantHTT aggregates in a mouse model.

Ji et al's degrader AUTOTAC is the second example of a degrader that degrades protein aggregates. Similar to AUTAC, it is a hetero-bifunctional degrader, which allows to design for a variety of substrates by the selection of a target-binding ligand (TBL). Their degradation tag binds to p62/sqstm1 and generates insoluble aggregates of p62. (I think this tag should not be called autophagy-targeting ligand: ATL, because p62 is not autophagy-specific; p62-ligand is more appropriate). In this case, the substrate bound to the target-binding ligand (TBL) crashes out with p62. This insoluble material is degraded by autophagy (it is not appropriate to describe this insoluble p62 aggregate as an autophagy-compatible form; Abstract, line 128, 141)

Although the AUTOTAC approach is unique and has potential, however, the manuscript requires significant strengthening to support the conclusion that AUTOTAC degrades cargos by "activating p62 and enhancing autophagic flux". Based on the UPS, so many degraders have been reported. Some of these also inhibit protein aggregates associated with neurodegenerative diseases such as mHTT (eg. S. Tomoshige et al., Angew. Chem. Int. Ed. 2017, 56, 11530). Perhaps, though, by degrading them at the monomer or oligomer stage. For this manuscript to satisfy Nature Commun readers, authors would need to provide clear evidence that AUTOTAC is different from the UPS-based degraders. Unfortunately, the current data do not sufficiently support the conclusion. I would like to discuss a few important issues here.

Note: In response to your valuable comments, we performed a series of additional experiments and made changes throughout the manuscript. The experimental results are now presented in the revised manuscript as **Figure 4b** and **Supplementary Figs. 2h, 2i, 2l, 4a, 4b, 4c, 4d, 5e, 5g, 5h, 6c, 6d, and 6h**.

Major comments

Comment 1. The authors argued the co-localization of LC3 with p62 as evidence for isolation membrane recruitment, but this is not convincing. p62 has a LIR motif that binds to LC3, so puncta that co-localize LC3 and p62 are not necessarily autophagosomes. It could be a membrane-free p62 aggregate or droplet. The WB experiments with DHQ are not fully convincing as an evidence for the autophagic mechanisms. The authors should examine if the lysosomal acidification of the AUTOTAC's target proteins. For example, it needs to be shown that hTau aggregates, which fused with pH-sensitive fluorescence, are acidified as a result of an AUTOTAC treatment. The presence of an isolation membrane around the aggregates can be most reliably verified using electron microscopy. Alternatively, it may be useful to analyze the localization of proteins that act during autophagosome formation, such as WIPI2.

Reply: To examine the lysosomal acidification of the target proteins of the AUTOTAC technology, we chose mutant tauP301L as a model substrate to create a mRFP-GFP-hTauP301L autophagy flux reporter (ptfTauP301L). Using this construct, we now show the lysosomal acidification of recombinant mutant mRFP-GFP-hTauP301L upon AUTOTAC treatment (**Supplementary Fig. 6h; page 14, lines 354-358**). In addition, we have also included data showing the co-localization of target proteins (ER β , MetAP2, FK2 and p62) with not only the autophagosome formation marker WIPI2 upon their respective AUTOTAC (**Supplementary Fig. 4a; page 8, lines 210-211**) (**Supplementary Fig. 5d; page 11, lines 291-293**) or ATL (**Supplementary Fig. 2h; page 6, lines 159-160**) treatment but also the lysosomal marker LAMP1 (**Supplementary Fig. 2i; page 6, lines 160-161**) (**Supplementary Fig. 5e; page 11, lines 293-294**) as we have previously reported in 19' *Mol. Cell.* 75:1058.

Comment 2. The authors do not distinguish the following mechanisms. "Aggregation-binding AUTOTAC" may recruit p62 to intracellular aggregates (e.g. Tau) for degradation as proposed in this manuscript. But it is also needed to consider that "aggregation-binding AUTOTAC" may newly form aggregates from p62 and soluble monomeric or oligomeric Tau. If the former mechanism exists, bilayered aggregates of Tau and p62 may be observed. The latter mechanism would also exist, if this is the case, colocalization data of LC3 with Tau-GFP alone (Fig. 5l) would not sufficiently support the model that AUTOTAC degrades protein aggregates. The reduction of Tau aggregates by AUTOTAC is nevertheless important, however, UPS-based degraders are also able to reduce protein aggregates by removal of aggregate-prone proteins at monomeric or oligomeric states.

Reply: We agree that AUTOTACs may either: 1) target inclusion bodies of already-aggregated proteins (e.g., tau) or 2) form new aggregates consisting of p62 and tau. Importantly, we feel that the latter would be a controlled form of aggregation (due to sequestration of target proteins and p62 complexes towards autophagic membranes). Thus, we speculate that either options 1 or 2 would result in degradation of the target protein, given that the self-polymeric propensity of p62 facilitates its autophagic targeting (i.e., mutant p62 lacking its ZZ domain and self-oligomeric tendency does not interact with LC3; 17' *Nat. Commun.* 8:102). Moreover, our data shows that AUTOTACs accelerate the elimination of high-molecular weight (Fig. 5l) and detergent-insoluble (Fig. 5m) tau species more than their monomeric (Fig. 5l) or detergent-soluble (Fig. 5m) counterparts. Finally, our data using the hTauP301L-BiFC murine model (Fig. 6c-h) shows that AUTOTAC selectively eliminates mutant hTauP301L (known to spontaneously misfold and aggregate) as opposed to endogenous murine wild-type tau. Taken together, our data supports our model that the PBA- and anle138b-based AUTOTACs degrade pre-existing or newly-formed protein aggregates. However, to verify our hypothesis, we have now included new data showing the lysosomal acidification of mRFP-GFP-hTauP301L inclusion bodies upon AUTOTAC treatment (**Supplementary Fig. 6h; page 14, lines 354-358**).

Comment 3. In Figure 4, the authors used the proteasome inhibitor MG132 to accumulate ubiquitinated proteins and examined the effect of AUTOTAC on the accumulated ubiquitin signal density. The title of Figure 4 includes "targeted delivery and degradation of misfolded protein cargos". It is quite misleading to describe as if all of the ubiquitinated proteins accumulated by MG132 treatment are misfolded. Ubiquitination is a posttranslational modification also involved also in signal transductions. The data in Fig. 4 in general do not support sufficiently the hypothesis that AUTOTAC delivered misfolded proteins to autophagy, and therefore requires revision.

Reply: The FK2 antibody used throughout Fig. 4 is specific only to proteins conjugated with poly- and/or mono-ubiquitin chains, and hence does not recognize free ubiquitin. Thus, identical to what we have done before in our previous literature (17' *Nat. Commun.* 8:102; 18' *PNAS* 111:E3853), we characterized the high-molecular weight FK2-stained bands as a model misfolding-prone, ubiquitinated and aggregated protein species. However, to better support our claim that PBA-based and anle138b-based AUTOTACs specifically deliver misfolded proteins and their high-molecular aggregates to autophagy, we have now included data that show selective degradation of mutant desminL385P (which is normally soluble and monomeric in its wild-type form, but highly UPS-resistant and aggregated once misfolded), a hallmark protein species in desminopathies, a model protein misfolding and aggregation disease (**Supplementary Fig. 5g,h; page 12, lines 1-18**).

Minor points

Point 1. Line 62: What does “autophagy-compatible form” means? There are no experimental data. If it means p62 aggregate, simply state it that way.

Reply: The phrase ‘autophagy-compatible form’ was used to describe the conformationally and functionally ‘activated’ sub-population of p62, whose ligand-bound ZZ domain results in ‘opening up’ of the PB1 domain away from the ZZ domain. Such ‘activation’ of p62 promotes its PB1 domain-dependent self-oligomerization and accelerates its targeting to autophagic membranes. Regarding this point, we have previously published experimental data supporting this claim in the following papers: 17' *Nat. Commun.* 8:102 and 18' *Nat. Commun.* 9:4373. To help the readers better understand this concept, we have revised the Introduction section (N-degron pathway paragraph) to definitively define the phrase “autophagy-compatible form” (**page 2, line 13; page 5, lines 297-314**).

Point 2. Line 69: “AUOTAC”

Reply: The typo has been corrected (**page 2, line 68**).

Point 3. Line 99: The cited reference 12 does not related to “neurodegenerative proteinopathies”. Please cite more appropriate reference here.

Reply: As keenly noted, an appropriate reference has been cited to support the claim that hallmark protein aggregates of neurodegenerative diseases are typically resistant to un/re-folding or proteasomal degradation, and that they even impair the proteasome (**page 3, lines 95-98**).

Point 4. This sentence is NOT sufficient to describe the recent progresses of autophagy-based degraders. The reference for AUTAC is cited but it does not explain what has been done such as removal of impaired mitochondria. Most importantly, the authors did not cite nor mention the ATTEC technology here (Nature, 2019). Moreover, a phrase of “targeting extracellular and secreted/membrane proteins directly to the lysosome (line 108) “is related to the LYTAC technology of Dr. Bertozzi but their work is not cited here (Banik, S. et al. Lysosome-targeting chimaeras for degradation of extracellular proteins. Nature <https://doi.org/10.1038/s41586-020-2545-9> (2020)).

Reply: As kindly suggested, we have now included all appropriate descriptions and references regarding the recent advances in autophagy/lysosome-based degraders or molecular glues in the Introduction section (**pages 3 and 4, lines 98-103**).

Point 5. Line 124: What are the criteria for “generally applicable chemical tool”? The criterion should be clearly indicated prior to the discussion if AUTOTAC meets this criterion or not. How about the previous techniques such as ATTEC?

Reply: We believe the criteria for ‘generally applicable chemical tool,’ in this case for degraders/molecular glues, is that any target protein should be relatively easily and efficiently degraded without too many technical limitations. While this definition may be too broad for practical use, we recognize that it may certainly apply to AUTAC, ATTEC and LYTAC (among others). As such, we have revised the sentence to now read “generally applicable TPD platform” (**page 5, lines 129-131**).

Point 6. Line 130: This reviewer understands that the ATL alone induces aggregation of p62. However, I do not understand why you suddenly asserted that AUTOTAC can induce the degradation of a broad range of substrates. I do not agree with the authors' notion that “ATL-mediated autophagic activation of p62 does not need to form a ternary complex”. In fact, the data in the following Results section seemed to suggest that AUTOTAC molecule binds with two proteins (substrate and p62) to form a ternary complex.

Reply: As astutely pointed out, p62 does indeed form a ternary complex with the AUTOTAC ligand and the target protein. We meant to say that unlike PROTAC, which relies on linker type- and exit vector-mediated positive cooperativity between the E3 ligase and the target protein for efficient degradation, AUTOTACs do not suffer from the same limitations. Indeed, the data from the original manuscript (Supplementary Fig. 6i) shows that the interaction between mutant tauP301L and mutant p62- Δ UBA, which cannot bind poly-Ub chains on cargoes, was still strengthened upon Anle138b-F105 treatment. To further support this claim, we have now added data in which a modified version of the tau degrader PBA-1105, with a drastically longer linker length, is still able to induce degradation of tau (**Supplementary Fig. 6c; page 13, lines 330-334**). As such, we have now revised the sentence to now read “As AUTOTACs do not require positive cooperativity between the target protein and p62 for efficient degradation of their ternary complexes, ...” (**page 5, lines 135-137**).

Point 7. Line 141: I do not understand clearly what is “autophagy compatible form”. It is “insoluble aggregate of p62”?

Reply: Please refer to our answer to specific point #1.

Point 8. Line 143: “YTK-F105, exhibited efficacy and selectivity to activate p62”

I could not find any data regarding the efficacy of these four compounds neither in Fig. 1b nor Suppl Fig. 1. The data in Fig. 1 showed that these compounds bind to p62 and induce p62 aggregation. But it does not mean that these compounds are selective to p62. I also do not understand what the authors mean by p62 activation. The data shows that the compounds induced aggregation of p62 and there is no other evidence that the aggregated p62 are “activated”.

Reply: The efficacy of the four ATL compounds in Fig. 1 are shown in the original manuscript by their interaction to p62 (Fig. 1c), efficacy of inducing p62 self-oligomerization (Fig. 1d), p62-LC3 puncta formation/co-localization (Fig. 1e, f, g), and induction of cellular autophagy flux (Fig. 1h, i.). To address the issue of specific/selective binding of the ATL to the p62 via its ZZ domain, we have now included data that show the interaction of ATLs with only p62 but not NBR1 (**Supplementary Fig. 2i; page 6, lines 167-169**). Since NBR1 carries a very similar ZZ domain and PB1 domain as p62 to similarly function as an autophagy cargo receptor, we feel that the ATLs' non-interaction with NBR1 supports their specific binding to p62. We have also revised the paragraph in question to better help the reader understand the p62-specificity of the ATL compounds. Regarding the ATL-induced activation of p62, please refer to our answer to specific point #1.

Point 9. Line 150: This pulldown assay showed YOK-1302 and YTK-1105 bind to p62. But specificity of these compounds to p62 are not shown in this manuscript. Considering the highly lipophilic structure of ATLs, this reviewer has a concern on their specificities to p62.

Reply: Please refer to our answer to specific point #8 regarding the selective binding of ATL to p62 but not NBR1, which carries a structurally and functionally similar PB1 and ZZ domain.

Point 10. Line 153-157: The colocalization of p62 and LC3 proteins should not be used as the sole evidence of autophagy. Because p62 has an LIR domain, it is known that p62 and LC3 often colocalize in punctate (aggregates or condensates) without an autophagic membrane (see Komatsu, Cell 2007, 131, 1149 & Pankiv, JBC 2007, 282, 24131). Therefore, the notions such as “(ATLs) target p62 to autophagosome membrane” or “(ATLs) facilitate autophagosome biogenesis# are not sufficiently supported by the data presented. To show that ATL induces autophagy (biogenesis of isolation membrane), experiments with cell lines that lack an autophagy-initiation complex is required. Deletion of Atg13 or FIP200 are frequently used to examine if a treatment induces autophagy. Mizushima reported a quantitative method to analyze the autophagy levels in culture cells and living animals (Kaizuka et al., Mol Cell, 2016, 64(4):835-849). The existence of membrane of autophagosome at the ATL-induced p62 aggregates may be examined with the colocalization of Syntaxin-17 or electron microscopy.

Reply: As kindly pointed out, we have now included data that show the efficacy of ATLs in inducing autophagy (biogenesis of isolation membrane) via the puncta formation and co-localization of p62 and WIPI2, which is a canonical and selective autophagosome biogenesis marker (**Supplementary Fig. 2h; page 6, lines 159-160**). In addition, we now show that ATLs target p62 to not only autophagic membranes but also the lysosome via co-localization analysis with the lysosomal membrane marker LAMP1 (**Supplementary Fig. 2i; page 6, lines 160-161**).

Point 11. Line 157-160: The authors rely heavily on an WB-based autophagic flux assay. This is not fully convincing. Please see the discussion on the major points above.

Reply: Please refer to our answers to major point #1 and specific point #10.

Point 12. Line 162-170: “We speculated....First, AUTOTAC brings a target to p62 via....Secondary,...” Some of these speculations are not examined experimentally in this manuscript

(e.g. catalytic action of AUTOTAC). Please focus on the topics you are going to examine in the following result section.

Reply: Our intentions with these speculations were to help the reader understand the AUTOTAC technology. We have also added new data supporting our speculations (e.g., catalytic nature of AUTOTACs) (**Supplementary Fig. 4h; page 9, lines 229-231**) (**Supplementary Fig. 6d; page 13, lines 337-338**). However, we recognize that some of the speculations are not experimentally examined and thus have revised the paragraph in question to better focus on the topics actually examined in the Results section.

Point 13. Line 171: “These mechanisms are independent of PPI... thus are generally applicable for a broad range of intracellular protein” It is too assertive. Please note not all of the speculated mechanisms are fully examined in the manuscript.

Reply: We think that the mode-of-action for AUTOTACs is indeed independent, or at least not as dependent as PROTACs, of protein-protein interactions between the target proteins and p62. The reason for our belief is that in the original manuscript, treatment with AUTOTAC resulted in the interaction between mutant hTauP301L-GFP and mutant p62 Δ UBA-myc (Supplementary Fig. 6j). Normally, the UBA domain of p62 is required for its interaction to poly-ubiquitinated tau, as is supported by both our data and known literature (05' *J. Neurochem.* 94: 192-203). Consistent with this, we observed that the interaction between mutant p62- Δ UBA and hTauP301L was strengthened upon AUTOTAC-treated conditions (Supplementary Fig. 6i). However, as we stated in our answer to comment #12, we have revised the paragraph in question to less overstate our speculations (**page 7, lines 175-185**).

Point 14. Line 179: “cells at (Fig. 2f-i).” Please remove “at”.

Reply: The grammatical error has been removed (**page 7, line 192**).

There are multiple errors in panels of Fig. 2 and I was not able to examine fully if the data support conclusion of the authors. Some of the errors include following 15) - 19).

Point 15. Line 178: “(Fig. 2d,e)”

I guess this should be corrected as “Fig. 2f,g”

Point 16. Line 179: “(Fig. 2f-i)”

I guess this should be corrected as “Fig. 2j,i”.

Point 17. Line 187: “Fig. 2j-l”

I guess this should be corrected as “Fig. 2k,l”

Point 18. Line 189: “Fig. 2m,n”

I guess this should be “Fig. 2d,e”.

Point 19. Line 190: “Fig. 2i,l,o”

This may be corrected as “Fig. 2n, m,o”.

Reply: We very much appreciate your constructive comments. We revised the manuscript in its entirety to make sure every panel is called correctly.

Point 20. Lines 190-191: “These results validate AUTOTAC as a general chemical tool for targeted proteolysis.” I was so confused by the errors in Figure 2 but it seems like levels of three protein targets

decreased in WB analyses. Is this the sole reason by which the authors concluded this is a “general” chemical tools?

Reply: We have now revised the sentence to read “These results validate AUTOTACs as robust degraders for targeted proteolysis of intracellular oncoproteins” (**page 8, lines 204-205**).

Point 21. Line 193: vinvlozolin-2204 induced the formation of AR+p62+ complexes and AR+LC3+ autophagic membranes (Fig. 3a,b). Fig. 3a and b showed colocalization of AR and LC3 but the data for colocalization with p62 is not presented. Moreover, the colocalization with LC3 signals does not always means the existence of “autophagic membrane”. LC3 colocalizes with p62 aggregates or droplets without “autophagic membrane”. To support the authors conclusion, existence of autophagic membrane must be examined by additional experiments. Thus, the conclusion is not sufficiently supported.

Reply: The typo has been corrected to now read “...formation of AR⁺LC3⁺ punctate structures” (**page 8, lines 206-207**). Please also refer to our answer to major point #1 for additional proof that AUTOTAC induces targeting of oncoproteins (e.g., ERbeta) to actual autophagic membranes marked with WIPI2 (**Supplementary Fig. 4a; page 8, lines 210-211**).

Point 22. Line 193-196: PHTPP-1304.....(Fig. 3c,d).

Dose-dependent formation of p62/ERB puncta was shown but this dots formation and colocalization ratio of LC3 (Fig. 3d) is not a convincing indicator of autophagy flux. To demonstrate the lysosomal degradation of the cargos, use of pH-sensitive protein probes such as GFP-RFP, Keima, or Rosella provides convincing results. To demonstrate the autophagy induction (increase of autophagy flux) during the treatment with AUTOTAC compounds, the use of Atg13 or FIP200 knockout cells (NOT knockdown) is recommended.

Reply: We show in the original manuscript that AUTOTAC-mediated degradation of estrogen receptor beta is dependent on both p62 and the macroautophagy marker ATG5 (Fig. 3f), which we believe corroborates our findings that AUTOTAC increases targeting of ERβ to p62-associated inclusion bodies. To confirm that these bodies are indeed autophagic membranes, we have now included data that show formation of ERβ⁺WIPI2⁺ punctate structures (**Supplementary Fig. 4a; page 8, lines 210-211**).

Point 23. Line 197. Fig. 3: MetAP2 level in the Fuma.-105 (+) & Baf A1 (+) lane is significantly higher than that in the Fuma.-105 (-) & Baf A1 (+) lane. Why?

Reply: We feel that the higher level of MetAP2 in the AUTOTAC+Baf.A1-treated lane when compared to that of only Baf.A1-treated lane is natural due to the increased autophagic flux of MetAP2 upon AUTOTAC treatment, as indicated by the normalized MetAP2 autophagic flux indices. This is most likely a result of the experimental condition (i.e., 200 nM and 6 h treatment of bafilomycin A1), which we feel is not enough to completely inhibit the entirety of the cellular autophagic flux.

Point 24. Line 199 Fig. 3f: Please add imaging data of the cells under p62 or Atg5 KO conditions to examine the numbers and characters of punctates.

Reply: As recommended, we have now included immunostaining data of the punctate structures of estrogen receptor beta in wild-type and ATG5 knock-out MEFs (**Supplementary Fig. 4b-d; page 8,**

lines 215-217). As expected, AUTOTAC treatment induced the punctate formation of MetAP2 or ER β only in wild-type mouse embryonic fibroblasts, but not in p62^{-/-} or ATG5^{-/-} cells.

Point 25. Line 229 Fig. 3k, l: These data are obtained with Fuma-105, however, the figure legends describe as “treated with PHTPP-1304, PHTPP”.

Reply: The typo has been corrected (**page 10, line 4**).

Point 26. Line 247: How the molecules “activated” p62? What does “activation” means?

Reply: Please refer to our answer to specific point #1.

Point 27. Line 248 (important): There are no evidence shown in the Fig. 4c regarding the coexistence of cargoes with oligomeric p62.

Reply: The sentence in question has been revised to now read “...triggered its self-oligomerization” (**page 10, line 250**).

Point 28. Line 250 Fig. 4d: The level of ubiquitinated proteins decreased in PBA1105(+) & HCQ(+) lane. Do you mean the decrease is not mediated by lysosomal degradation?

Reply: Since the normalized autophagic flux index of high-molecular ubiquitinated protein species (stained by FK2) is higher by approximately two-fold upon AUTOTAC treatment, we feel that PBA-1105 induces the autophagic degradation of said species. As for why the levels of FK2 are lower in PBA1105⁺HCQ⁺ lane as compared to HCQ⁺ lane, we speculate this may be due to our experimental conditions in which HCQ (5 μ M, 24 hours) was not sufficient for complete inhibition of the cellular autophagic flux (as a comparison, most chloroquine or HCQ treatment is carried out at higher concentrations by other groups). In this case, treatment with AUTOTAC would lead to a decrease in both HCQ-untreated and -treated samples, but still increase the autophagic flux of FK2.

Point 29. Line 253: Please state the number of p62-dots before and after the AUTOTAC treatments. This should be an important factor whenever you discuss the colocalization ratio of something with p62 dots.

Reply: In our experimental conditions upon PBA-1106 AUTOTAC treatment (1 μ M, 24 h), we did not notice a statistically significant increase in the number of p62 punctate structures compared to control samples, as is evident in Fig. 4h (first row). However, what we did notice was that under HCQ-treated conditions, PBA-1106 AUTOTAC co-treatment generated the most number of p62 punctate structures (average of 14 ± 4.2 p62 puncta per cell) compared to ATL or TBL alone. We have revised the section in question to better reflect this (**page 11, lines 278-281**).

Point 30. Line 254: Existence of autophagic membrane is not shown with experimental evidence.

Reply: Please refer to our answers to major point #1 and to specific point #10. We have also revised the sentence to now read “...p62⁺LC3⁺ autophagic membranes” (**page 11, lines 277-278**).

Point 31. Line 256: “UPS-resistant misfolded protein aggregates” are not appropriately analyzed in Fig.4. For example, there are no data regarding the numbers of aggregates. Moreover, the authors need to consider aggregates may be generated by the action of AUTOTAC via p62 destabilization.

Reply: To strengthen our characterization of “UPS-resistant misfolded protein aggregates,” we now show data in which AUTOTACs selectively degrade mutant, aggregation-prone desminL385P as opposed to wild-type desmin (**Supplementary Fig. 5g,h; page 12, lines 297-314**). While wild-type desmin does not misfold nor aggregate and is diffusive throughout the cytosol, mutant desminL385P spontaneously aggregates into well-defined punctate structures and is resistant to UPS (Liu *et al.*, 2006. *FASEB. J.*, **20**: 362-364). Regarding the reviewer’s point on AUTOTAC-mediated aggregate formation, we believe that this is a controlled form of aggregation as a result of sequestration of target proteins with p62 complexes (please refer to our answer to major point #2). Thus, this mode-of-action would not be pathological since otherwise aggregated and detergent-insoluble protein species (e.g., Ub-conjugated protein aggregates, mutant desmin and mutant tau) would not have been degraded as we show in Fig. 5.

Point 32. Line 263: The level of ubiquitinated proteins decreased in MG132(+), Anie-F105(+) & HCQ(+) lane. Do you mean the decrease is not mediated either by lysosomal degradation or proteasomal degradation?

Reply: Please refer to our answer to specific point #28, but with Anle138b-F105 instead of PBA-1105.

Point 33. Line 264 Fig 4k: The increase of FK2 colocalization with p62 alone does not sufficiently support the conclusion that the compounds work via “p62-dependent macroautophagy”.

Reply: We have revised the sentence to now read “...via p62-associated macroautophagy.” Moreover, we feel that our new data indicating the punctate formation and co-localization of Ub-conjugated protein aggregates with either WIPI2 (**Supplementary Fig. 5d; page 11, lines 290-293**) or LAMP1 (**Supplementary Fig. 5e; page 11, lines 293-294**) suggest p62-associated macroautophagy is at work.

Point 34. Line 277 Supplemental Fig. 5c,d: The data do not exclude a possibility that 4-PBA may promiscuously bind to proteins.

Reply: Given its nature as a chemical chaperone, we agree with the reviewer that the interactome of 4-PBA may be relatively large. However, our initial decision to create an AUTOTAC chimeric degrader with a chemical chaperone moiety was to test the proof-of-concept that misfolded and/or aggregation-prone protein species could be targeted to p62-dependent macroautophagy. For future studies, we plan to create AUTOTACs with a target-binding moiety as specific to one protein species as possible.

Point 35. Line 288: This colocalization analysis alone is not sufficient to support their conclusion that PBA-1105 AUTOTAC selectively induced the sequestration and autophagic targeting of tauP301L.

Reply: Please refer to our answer to major point #1.

Point 36. Line 308: To prove that the effect of AUTOTAC on aggregate-prone protein is independent of polyubiquitin, it is better to use E1 inhibitor (e.g. pyr41).

Reply: We feel that our data in which mutant p62 lacking the UBA domain can bind the otherwise non-interactable tau upon AUTOTAC treatment (Supplementary Fig. 6j), along with the ubiquitin-independent degradation of oncoproteins (Fig. 3g,h and Supplementary Fig. 4f) is sufficient to establish the ubiquitin-independent mode-of-action for AUTOTACs. Moreover, we feel that the low efficacy of pyr41 ($IC_{50} < 10 \mu M$) and its side effect of up-regulating global sumoylation pose significant limitations in its usage.

Point 37. Lysosomal acidification of Tau aggregates must be monitored with pH-sensitive fluorescent probes.

Reply: Please refer to our answer to major point #1.

Point 38. Line 313: Macroautophagy works only at cytoplasm. How could AUTOTACs degrade Htt-NLS-GFP?

Reply: We speculate that the presence of nuclear subpopulations of p62 and LC3, which has been reported in the literature, allow AUTOTACs to target nucleus-resident mutant HttQ97 for autophagic degradation (perhaps as a form of macronucleophagy).

Point 39. Line 325: Supplemental Fig. 6k,l: Colocalization of cargo with LC3 is not the conclusive evidence that AUTOTAC selectively promoted autophagic targeting of Htt.

Reply: Please refer to our answer to major point #1. We feel that our new data showing: 1) the targeting of p62 and target proteins to autophagic membranes by AUTOTAC/ATLs and 2) the lysosomal acidification of mutant tau by the same degraders suggest that these AUTOTACs selectively promoted autophagic targeting/lysosomal delivery of cargoes, since the mechanism-of-action by which AUTOTAC delivers substrates should be identical for mutant tau and huntingtin.

Point 40. Line 339: The selective reduction of hTau over murine WT Tau is remarkable. Please add the data showing the compound selectively bind hTau over murine WT Tau.

Reply: We believe that PBA-based and anle138b-based AUTOTAC compounds discriminate between wild-type and mutant tau species by their tendency to misfold and aggregate, as evidenced by our *in vivo* murine data showing selective degradation of recombinant mutant hTauP301L as opposed to endogenous wild-type murine tau. However, since wild-type tau is also prone to spontaneous misfolding and aggregation (especially due to tau 'seeds'), we feel that technical constraints limit us from conclusively showing that our compound binds only hTau over murine WT tau. Indeed, our own data in the original manuscript (Fig. 6d,h) show that at high concentrations, PBA-1105 injection may also decrease insoluble levels of mTau (along with hTau).

Point 41. Line 408: "It is highly likely..." It is too speculative.

Reply: We have now included data that show AUTOTACs are recycled from the lysosome for subsequent rounds of degradation. Specifically, under conditions of proteasomal inhibition, PHTPP-1304 AUTOTAC exhibited degradative efficacy against estrogen receptor beta of up to at least 8 hours

post-washout (**Supplementary Fig. 4h; page 9, lines 229-231**) Similarly, Anle138b-F105-induced degradation of mutant tauP301L persisted up to at least 8 hours post-washout (**Supplementary Fig. 6d; page 13, lines 337-338**), indicating the catalytic nature of AUTOTACs.

REVIEWER 3

Remarks to the Author: In this manuscript, Ji et al. developed a new technology, termed AUTOphagy-Targeting Chimera (AUTOTAC), as a chemical tool that can degrade specific proteins by autophagy. AUTOTAC is an artificial molecule composed of two parts: (1) target-binding ligand that interacts with specific substrates, and (2) autophagy-targeting ligand that binds with the ZZ domain of p62, a selective substrate of autophagy. The authors developed multiple AUTOTACs targeting several oncoproteins and degradation-resistant aggregates and showed that these AUTOTACs could induce degradation of these targets in culture cells and in the brain. This study provides a potentially useful tool that can compensate the previously established other targeted protein degradation technologies (e.g., PROTAC and AUTAC). Overall, the experiments are well performed, but there are some concerns about the effects of AUTOTACs on autophagy and p62.

Note: In response to your valuable comments, we performed a series of additional experiments and made changes throughout the manuscript. The revised manuscript now describes these revisions (**Supplementary Figs. 2j, 2k, 2l, 2m, 4g, 4h, 5c, and 5f**) along with appropriate changes to the text.

Major comments

Comment 1. In this manuscript, the effect of autophagy-targeting ligands on autophagy flux is not clearly demonstrated. In Fig. 1h-k, the authors showed that YOK1304 and YTK105 increased the autophagy flux using a lysosomal inhibitor hydroxychloroquine (HCQ). However, the results are not convincing because the amount of p62 and LC3-II under HCQ treatment increased more in YOK1304- and YTK105-treated cells than in control cells. Such changes could result from enhanced transcription, translation and/or protein stability (e.g., by suppression of proteasomal degradation) of p62 upon treatment with YOK1304 and YTK105. To convincingly support their hypothesis that YOK1304 and YTK105 enhance the degradation of p62 by autophagy but not proteasomal degradation, the authors should perform experiments using autophagy-deficient cells and proteasomal inhibitors as the authors have previously performed similar experiments for other ligands (Cha-Molstad et al., Nat Cell Biol. 2015). In these experiments (including autophagy flux assays in Fig. 1h-k), the authors should include YT-8-8 and YOK-2204, latter of which is used in later parts of this study but not included in Fig. 1h-k, and quantitatively and statistically analyze the results.

Reply: We believe that the level of p62 and of LC3-II under HCQ treatment is higher with YOK1304- and YTK105-treated cells than in control cells because in our experimental conditions, HCQ (10 μ M, 24 h) is not enough to completely inhibit all of the cellular autophagy flux. Hence, induction of autophagy flux using the ATLS would then result in greater amounts of p62 and LC3-II being available for accumulation upon flux inhibition by HCQ. However, as kindly recommended, we have now included data that show 1) the autophagy flux indices for YOK-2204 (**Supplementary Fig. 2j, k; page 6, lines 163-167**) and 2) proteasome-independent effect of YOK-2204 (**Supplementary Fig. 2k; page 6, lines 166-167**). Moreover, with AUTOTACs, we also now show that the AUTOTAC-mediated degradation of ER β (**Supplementary Fig. 4g, h; page 9, lines 225-233**) or Ub-conjugated protein aggregates (**Supplementary Fig. 5c,f; page 11, lines 282-284 and 294-296**) is proteasome-independent but macroautophagy- and p62-dependent.

Comment 2. Related to above concern, in Fig. 3f, the authors showed that knockdown of p62 and

ATG5 abolished the PHTPP-1304-induced degradation of ER β . However, the results are confusing because the amount of ER β strongly increased in p62- and ATG5-depleted cells even without PHTPP-1304 treatment. The authors should quantitatively and statistically analyze the results and discuss the possible causes of this phenomenon. The authors should also perform similar experiments using p62- and ATG5-depleted cells in other ligand-substrate pairs (e.g., Fuma.-105-MetAP2, Vnc-2204-AR, and PBA-1105-Tau) to investigate whether these substrates are indeed degraded dependently on p62 and autophagy.

Reply: We feel that ER β levels were significantly up-regulated upon siRNA-mediated interference of p62 and ATG5 because endogenous ER β is degraded via macroautophagy. This is in agreement with other published findings in the literature that show the autophagic degradation of nuclear receptors, including estrogen receptor alpha androgen receptor (15' *Cell Signal.* 27:1994; 14' *PloS one.* e94880; 11' *J Biol Chem.* 286: 22441). Consistently, we also now show data in which up-regulated levels of estrogen receptor beta upon proteasomal inhibition are efficiently eliminated by PHTPP-1304 (**Supplementary Fig. 4g, h; page 9, lines 225-233**). Moreover, as requested, we now show data in which the degradation of Ub-conjugated protein aggregates (FK2) by AUTOTACs is indeed dependent on both p62 and ATG7 (**Supplementary Fig. 5c,f; page 11, lines 282-284 and 294-296**).

Comment 3. The authors hypothesized that autophagy-targeting ligands interact with the ZZ domain of p62. It can be experimentally tested using p62 mutant lacking the ZZ domain in Fig. 1c.

Reply: In our previous study (17' *Nat. Commun.* 8:102), we have shown that autophagy-targeting ligands bind to p62 via its ZZ domain by virtue of its D129 and CXXC zinc finger motifs. We also now show in this paper that the optimized ATLs bind to only wild-type p62 (Fig. 1c and **Supplementary Fig. 2l; page 6, lines 167-169**), but not its mutant D129A counterpart lacking a critical N-degron recognition residue (**Supplementary Fig. 2m; page 6, lines 170-172**) nor NBR1, which carries a very similar ZZ domain and is also an autophagy cargo receptor (**Supplementary Fig. 2l; page 6, lines 167-169**)

Minor comments:

There are many mistakes in the text, figures, and figure legends:

P6, line 151, YOK-1304 is not shown in Fig. 1c.

P6, line 151, YTK-1105 is not shown in Fig. 1b.

P7 line 178, "Fig. 2d, e" should be "Fig. 2f, g".

P7 line 179, "Fig. 2f-i" should be "Fig. 2i, j".

P7 line 180, "Fig. 2d, e" should be "Fig. 2f, g".

P7, line 181, "Fig. 2h, i" should be "Fig. 2n, p".

P7, line 187, "Fig. 2j-l" should be "Fig. 2k, l".

P7, line 189, "Fig. 2m, n" should be "Fig. 2d, e".

P7, line 189, "Fig. 2n" should be "Fig. 2h".

P7, line 190, "Fig. 2i, l, o" should be "Fig. 2e, m, p".

P7, line 193, there is no data corresponding to "the formation of AR+p62+ complexes".

P8, line 202, "Supplementary Fig. 4a" should be .

P9, line 220, Fig. 3J is not cited.

P10, line 253, “Fig. 4h, k” should be “Fig. 4h, i”.

P11, line 285, “Fig. 5j” should be “Fig. 5i, j”.

P12, line 298, “6 and 8” should be “7 and 8”.

P12, line 315, “Supplementary Fig. 6c-d” should be “Supplementary Fig. 6a, b”.

P12, line 317, “Supplementary Fig. 6e” should be “Supplementary Fig. 6c”.

P12, line 318, “Supplementary Fig. 6a, b” should be “Supplementary Fig. 6d, e”.

P13, line 320, “Supplementary Fig. 6b, d” should be “Supplementary Fig. 6b, e”.

P43, line 975, “YOK-1104” should be “YOK-1106”.

P52, line 1040, “YOK-1104” should be “YOK-1106”.

P53, line 1054, “YOK-1104” should be “YOK-1106”.

Supplementary Fig. 4b is not cited in any parts of this manuscript.

Reply: We very much appreciate these constructive comments. All these comments were editorially addressed in the revised manuscript and double-checked to make sure every figure is called correctly in both the main text and in the figure legends.

REVIEWER COMMENTS

Reviewer #1 (Remarks to the Author):

The authors have addressed all of my concerns. This manuscript can be accepted with some minor revisions.

Regarding “ATTEC is a molecular glue specific only to mutant Htt proteins”, the Lu group recently showed that ATTEC ligand can be conjugated with a lipid droplet target ligand to degrade lipid droplets via autophagy. <https://www.nature.com/articles/s41422-021-00532-7>

In Fig.1c, it was mentioned that “biotinylated ATLS confirmed that p62 bound YT-8-8, 154 YOK-2204, and YTK-105”. The structures of the biotinylated ATLS should be included in the SI.

Reviewer #2 (Remarks to the Author):

Remarks to the Author

The revised manuscript by Ji et al., has eliminated many typographical and other errors that were included in the original manuscript.

As the authors have responded to other reviewers' comments, this manuscript can be described as a study for Proof-of-concept. Therefore, I pay attention to whether the mechanism proposed by the authors is well supported.

Major comment 1.

The authors should change the name of their degraders to something other than “autophagy targeting chimera”.

"Autophagy targeting chimera" was actually defined as the (unabbreviated) formal name of the AUTAC technology in a published paper (D. Takahashi et al. Mol Cell, 2019).

Therefore, a different name should be used for the technology described in the current paper; AUTACs and the authors' "AUTOTACs" are different technologies, and there are serious concerns that giving a completely identical name to different technologies will lead to confusion.

"Autophagy targeting chimera" has been abbreviated to AUTACs in the previous papers. The use of different abbreviations (AUTAC and AUTOTAC) for the same name will also cause serious confusion.

Major comment 2.

The authors should be more rigorous in their use of terminology." The phrase "Autophagy-targeting ligand (ATL)" appears frequently throughout the paper. This study uses a compound that binds to the p62 protein as the "ATL" ligand. Why not use the more accurate term: p62 ligand?

The p62 protein is not a component of the core autophagy machinery. Therefore, p62 binders should not be referred to as ATL. I think it is important to use the scientifically accurate terminology.

Major comment 3. (Regarding my previous major comment 2)

I previously suggested an experiment with hTau aggregates, which are fused with pH-sensitive fluorescence protein. If the authors are able to show acidification of the preexisting aggregates as a function of an AUTOTAC treatment, this will greatly enhance the importance and impact of this study.

The authors added new data related to this point (Suppl Fig 6h). I think the quality of the image is not sufficient to support the revised sentences (lines 354-358). In an acidic environment (lysosome), the fluorescence of GFP must decrease against that of RFP. Unfortunately, The images are not good enough, and I do not see such a decrease of GFP fluorescence in the added Suppl Fig. 6h. The data in Suppl Fig. 6h also need quantifications.

At least, the increase or decrease in the number of aggregates and increase of the number of aggregates under an acidic environment should be assessed in the presence/absence of AUTOTAC.

It is not convincing that AUTOTAC was able to target preexisted aggregates to autophagy and therefore the manuscript needs additional experiments.

In a previous response to Major comment 2, the author agreed with me that the two mechanisms can coexist. 1) AUTOTAC degrades aggregates already present in the cell, and 2) AUTOTAC-induced degradation of new aggregates of p62 and substrates.

Of these, mechanism 1) is very interesting, but as discussed above, the data in the current revised manuscript do not provide sufficient evidence for the existence of this mechanism. Additional experiments are needed.

Soluble proteins can be degraded by mechanism 2) once they form aggregates, as there are examples in this manuscript of the degradation of several proteins such as MetAP2.

The mechanism in (2) increases the number of aggregates in the cell, which can potentially have unwanted effects.

The above possible mechanisms are not well described in the revised main text. Please consider further revisions.

Specific comment 1.

page 4, line 4

Following paper describes the application of ATTEC technology to degradation of substrates other than mHtt.

Fu Y, Chen N, Wang Z, Luo S, Ding Y, Lu B. Degradation of lipid droplets by chimeric autophagy-tethering compounds. Cell Res. 2021 Jul 8. doi: 10.1038/s41422-021-00532-7. Epub ahead of print. PMID: 34239073.

Specific comment 2.

page 4, line 11 “no degrader technology yet exists for directly targeting proteins and their aggregates to autophagosome in an Ub-independent and receptor mediated manner”.

This is not an accurate description of the existing technologies: ATTEC binds substrates to LC3 proteins located in the autophagosome membrane. In other words, ATTEC is a direct approach between substrate and autophagosome, and it is a ubiquitin-independent mechanism. Although p62 is a preferred substrate for autophagy, AUTOTAC does not direct the substrate directly to the autophagosome.

Specific comment 3. (Regarding my previous specific comment 8)

page 4 line 167 and Suppl Fig. 2I

The added data (Suppl Fig.2I) shows the difference in binding properties for p62 and NBR1 to ATL ligands. However, This is not the point I wanted to address. This is not sufficient for the purpose of determining whether ATL binds to other proteins (not limited to known autophagy receptors). A

pull-down experiment using ATL may be easily performed, and LC-MS can analyze ATL interactors other than p62.

This comment is important because it is regarding the selectivity of AUTOTACs and its possible side effects.

Specific comment 4.

page 5, line 135

I understood your discussion on my previous specific comment 6. However, the importance of positive cooperativity is so far known only for PROTACs, and these backgrounds does not appear before this sentence. I feel that the sudden mention of "positive cooperativity" may confuse the reader.

Note that in this revised manuscript, the author does not provide conclusive evidence that "positive cooperativity" is not important. There is only an example of a compound with a long linker, which does not convincingly argue for the general importance/non-importance of cooperativity in AUTOTAC experiments.

Specific comment 5.

page 6, line 154

Please provide the chemical structures of biotinylated ATLs in the figures.

Specific comment 6.

page 6, line 159

WIPI2 and LAMP1 localize at different steps in the autophagy processes. The colocalization data in Suppl Fig.2 are not quantified and do not provide any time course-related information. Overall, the revisions made to show the recruitment of autophagosome membrane to the p62 aggregates do not sufficiently address the concerns in my previous comments (major comments 1).

Specific comment 7.

page 7, lines 175-185. (Regarding my previous specific comments 12 and 13)

These sentences are speculations without sufficient support of experimental evidence. I would like to suggest removing these from the Result section and including them in the Discussion section.

Specific comment 8.

page 9, line 229

The observation that the level of substrate did not increase 8 hours after AUTOTAC wash out is not reliable evidence to discuss whether AUTOTAC has catalytic properties.

Specific comment 9.

page 11, line 282 and Suppl Fig. 5c

The levels of Atg7 protein were not sufficiently decreased by Atg7 siRNA treatment. This experiment is not trustworthy. The conclusion that AUTOTAC is autophagy-dependent cannot be validated by this data. Please also note that bands of LC3-II appear in the 2 lanes in the WB image where the author claimed the Atg7 knock-down. If Atg7-KD was successful, autophagy activity would decrease so that LC3-II bands are expected to disappear.

To obtain conclusive evidence that AUTOTAC is indeed autophagy-dependent, I would like to suggest avoid using RNAi of Atg p7 and use Atg KO cell lines. The authors used Atg5 KO MEF cells in a Suppl Fig. Please replace all the experiments using Atg7 RNAi with the data using Atg5 KO cells.

Specific comment 10 (Regarding my previous specific comment 31)

page 12, lines 297-314

I could not understand the author's argument about WT and mutant desmin well.

In this paragraph, it is discussed that mutant desmin, which is prone to aggregation, can be degraded by AUTOTAC, but wt-desmin is not. I was a little confused because the authors also state in this paper that some soluble proteins such as MetAP2 can be degraded by AUTOTACs.

ATL can aggregate p62, but do you mean this ability alone is not sufficient? Does this mean that the substrate must have aggregate-prone nature for successful AUTOTAC degradation?

Specific comment 11.

page 13, line 330 and line 339

Here, using just one compound, the authors conclude that linker length does not affect AUTOTAC-mediated degradation. This should be judged carefully. Without several compounds with different length linkers, no accurate conclusion can be drawn.

The efficiency of degradation is also affected by the magnitude of the interaction between TBL and the substrate. If the affinity of the TBL-substrate is small, the cooperativity of p62 with the substrate in the ternary complex may play a role.

I think the current description is a bit overstated.

Page 13, line 339

As I wrote as specific comment 8, The observation that the level of mutant Tau did not increase 8 hours after wash-out of AUTOTAC is not fully reliable evidence to discuss whether AUTOTAC has catalytic properties.

The revised sentence "These data validate catalytic efficacy of AUTOTACs" needs to be reconsidered.

Specific comment 12 (Regarding previous specific comment 38)

Macroautophagy proceeds only in cytoplasm so that substrate with nuclear-localizing signal (NLS) are expected to escape from degradation but here you found the successful degradation of a protein with NLS.

The authors' reply to my previous comment is below.

"We speculate that the presence of nuclear subpopulations of p62 and LC3, which has been reported in the literature, allow AUTOTACs to target nucleus-resident mutant HttQ97 for autophagic degradation"

It does not make sense to me. Nuclear localization of p62 and LC3 does not show the formation of isolation membrane in nucleus, because the isolation membrane originates from ER-mitochondria contact site outside of nucleus.

The result that proteins with NLS can be degraded makes me a little worried about the correctness of the mechanism depicted by the author in Fig. 7.

There is no description of why AUTOTAC can degrade nuclear protein in the revised main text. Please include your comments in the main text.

Specific comment 13.

page 107 Fig. 7

Figure 7 contains both the results and speculations obtained by this study. It is difficult for the reader to distinguish between the two, and there is a concern that this may be misleading.

For example, that AUTOTAC is recycled is a hypothesis, not directly proven by the data in this paper. As already pointed out, the wording p62 ligand should be used instead of ATL.

Proof of substrate transfer to the autolysosome can be examined with substrates that have pH-sensitive fluorescence, but the data in Suppl. Fig. 6h of this manuscript is of insufficient quality.

Changing the title of the figure to something like the Speculative model of AUTOTAC mechanism could be considered.

It is also difficult to understand from Fig. 7 how preexisted intracellular aggregates are degraded by the AUTOTAC mechanism. Could you draw a model in which p62 is newly recruited by the action of AUTOTAC around aggregates such as Tau? Is it possible to do additional experiments to support such a hypothetical diagram?

Specific comment 14.

page 109, Suupl. Fig. 2

Chemical structure of biotinylated YOK-2204 should be included.

The title of the figure starts with "Computational modeling of" but the figure include data unrelated to the computational studies.

Specific Comment 15.

page 111 legends for b and c.

"ATG-/-" Is it ATG5-/- MEF cells?

Reviewer #3 (Remarks to the Author):

The authors have addressed all the criticisms raised by this reviewer.

REVIEWER 1

Remarks to the Author: The authors have addressed all of my concerns. This manuscript can be accepted with some minor revisions.

Comment 1. Regarding “ATTEC is a molecular glue specific only to mutant Htt proteins”, the Lu group recently showed that ATTEC ligand can be conjugated with a lipid droplet target ligand to degrade lipid droplets via autophagy. <https://www.nature.com/articles/s41422-021-00532-7>

Reply: We have now revised the manuscript to cite and describe the development of lipid droplet-specific ATTEC degrader by the Lu group (lines 101-103).

Comment 2. In Fig.1c, it was mentioned that “biotinylated ATLS confirmed that p62 bound YT-8-8, YOK-2204, and YTK-105”. The structures of the biotinylated ATLS should be included in the SI.

Reply: The structures of the biotinylated ATLS are now included in **Supplementary Fig. 1e** (lines 145 and 155).

REVIEWER 2

Remarks to the Author: The revised manuscript by Ji et al., has eliminated many typographical and other errors that were included in the original manuscript. As the authors have responded to other reviewers' comments, this manuscript can be described as a study for Proof-of-concept. Therefore, I pay attention to whether the mechanism proposed by the authors is well supported.

Major comments

Comment 1. The authors should change the name of their degraders to something other than “autophagy targeting chimera”. “Autophagy targeting chimera” was actually defined as the (unabbreviated) formal name of the AUTAC technology in a published paper (D. Takahashi et al. Mol Cell, 2019). Therefore, a different name should be used for the technology described in the current paper; AUTACs and the authors' “AUTOTACs” are different technologies, and there are serious concerns that giving a completely identical name to different technologies will lead to confusion. “Autophagy targeting chimera” has been abbreviated to AUTACs in the previous papers. The use of different abbreviations (AUTAC and AUTOTAC) for the same name will also cause serious confusion.

Reply: The term “AUTOTAC (AUTOPhagy-TARGETing Chimera)” was filed for patent (US 62/702,473; provisional application, United States; 2018.07.24) and copyrights as the logo and name of the proprietary technology belonging to AUTOTAC Bio Inc. in conjunction with Seoul National University in 2018, prior to the publication of 19' *Molecular Cell* by Takahashi et al. Given our intellectual properties on AUTOTAC, we would be grateful if we could retain the term AUTOTAC in this paper. Additionally, we feel that the most usual source of confusion regarding an abbreviation is when one acronym is used for two or more different phrases, and not two distinct, copyrighted/patented acronyms for one term since an

acronym is used far more often than its full form.

Comment 2. The authors should be more rigorous in their use of terminology." The phrase "Autophagy-targeting ligand (ATL)" appears frequently throughout the paper. This study uses a compound that binds to the p62 protein as the "ATL" ligand. Why not use the more accurate term: p62 ligand? The p62 protein is not a component of the core autophagy machinery. Therefore, p62 binders should not be referred to as ATL. I think it is important to use the scientifically accurate terminology.

Reply: Extensive studies show that the archetypal autophagy cargo receptor p62 is involved in the degradation of an extremely long list of cargoes via macroautophagy. While it may not be a core autophagy-related (ATG) protein *per se*, we believe its ubiquity in cargo recognition and delivery is of utmost importance in macroautophagy. Moreover, as a component of AUTOTAC, we have patented and copyrighted in 2018 the phrase "autophagy-targeting ligand (ATL)" for small molecule ligands to p62 and other autophagy receptors (US 62/702,473; provisional application, United States; 2018.07.24). Thus, we would like to respectfully ask that the term ATL be left as it is in this paper. However, to accurately describe this and preclude any scientific misunderstanding, we have now included both this point in the first part of our Results section and mentions of 'p62-targeting ligands' and 'p62-binding moieties' in lieu of autophagy-targeting ligands/ATLs as much as possible.

Comment 3. I previously suggested an experiment with hTau aggregates, which are fused with pH-sensitive fluorescence protein. If the authors are able to show acidification of the preexisting aggregates as a function of an AUTOTAC treatment, this will greatly enhance the importance and impact of this study. The authors added new data related to this point (Suppl Fig 6h). I think the quality of the image is not sufficient to support the revised sentences (lines 354-358). In an acidic environment (lysosome), the fluorescence of GFP must decrease against that of RFP. Unfortunately, The images are not good enough, and I do not see such a decrease of GFP fluorescence in the added Suppl Fig. 6h. The data in Suppl Fig. 6h also need quantifications. At least, the increase or decrease in the number of aggregates and increase of the number of aggregates under an acidic environment should be assessed in the presence/absence of AUTOTAC.

It is not convincing that AUTOTAC was able to target preexisted aggregates to autophagy and therefore the manuscript needs additional experiments. In a previous response to Major comment 2, the author agreed with me that the two mechanisms can coexist. 1) AUTOTAC degrades aggregates already present in the cell, and 2) AUTOTAC-induced degradation of new aggregates of p62 and substrates. Of these, mechanism 1) is very interesting, but as discussed above, the data in the current revised manuscript do not provide sufficient evidence for the existence of this mechanism. Additional experiments are needed. Soluble proteins can be degraded by mechanism 2) once they form aggregates, as there are examples in this manuscript of the degradation of several proteins such as MetAP2. The mechanism in (2) increases the number of aggregates in the cell, which can potentially have unwanted effects.

Reply: We believe that the data in question using recombinant mRFP-GFP-hTauP301L accurately represents lysosomal acidification of pre-existing aggregates, because the tau inclusion bodies were formed via okadaic acid-mediated hyperphosphorylation and aggregation *prior* to the addition of the AUTOTAC compound. This was done, in contrast to existing data in the original manuscript where okadaic acid and AUTOTAC were treated in

combination, at the behest of Reviewer 2's first-round revision major comment. We apologize for the confusion that the original figure legend may have caused, and have now properly described the experiment (**Supplementary Fig. 6h-i**; lines 348-351). We have replaced the [okadaic acid + PBA-1105] image (obtained from the same experiment) to better show quenched tau aggregates, and have also now quantified the relative number and GFP/RFP signal of pre-formed tau aggregates in said experiment. Consequently, we believe that our new data proves that AUTOTACs are capable of targeting pre-formed aggregates. While this may not rule out the effect of forming new aggregates upon AUTOTAC treatment via the self-oligomeric tendency of p62, we feel that this would not only be controlled but a temporary sequestration step prior to degradation.

Specific comments

The above possible mechanisms are not well described in the revised main text. Please consider further revisions.

Comment 1. page 4, line 4. Following paper describes the application of ATTEC technology to degradation of substrates other than mHtt. Fu Y, Chen N, Wang Z, Luo S, Ding Y, Lu B. Degradation of lipid droplets by chimeric autophagy-tethering compounds. Cell Res. 2021 Jul 8. doi: 10.1038/s41422-021-00532-7. Epub ahead of print. PMID: 34239073.

Reply: The manuscript was revised to cite and describe the development of lipid droplet-specific ATTEC degrader by the Lu group (lines 101-103).

Comment 2. page 4, line 11 “no degrader technology yet exists for directly targeting proteins and their aggregates to autophagosome in an Ub-independent and receptor mediated manner”. This is not an accurate description of the existing technologies: ATTEC binds substrates to LC3 proteins located in the autophagosome membrane. In other words, ATTEC is a direct approach between substrate and autophagosome, and it is an ubiquitin-independent mechanism. Although p62 is a preferred substrate for autophagy, AUTOTAC does not direct the substrate directly to the autophagosome.

Reply: To properly distinguish existing degrader technologies from AUTOTAC, we have now revised the sentence to read “no degrader technology yet exists for directly sequestering and targeting proteins and their aggregates using an autophagy cargo adaptor (e.g., p62)” (lines 111-116).

Comment 3. (Regarding my precious specific comment 8) page 4 line 167 and Suppl Fig. 2l. The added data (Suppl Fig.2l) shows the difference in binding properties for p62 and NBR1 to ATL ligands. However, This is not the point I wanted to address. This is not sufficient for the purpose of determining whether ATL binds to other proteins (not limited to known autophagy receptors). A pull-down experiment using ATL may be easily performed, and LC-MS can analyze ATL interactors other than p62. This comment is important because it is regarding the selectivity of AUTOTACs and its possible side effects.

Reply: To definitively determine the off-target interaction of AUTOTACs and ATLs to proteins other than p62, we carried out a competitive radioligand binding assay to determine the inhibition of specific binding or activity of a wide variety and large number of receptors. The 68 tested receptors include those for adenosine A₁, adrenergic alpha/beta, calcium

channel N-type, cannabinoid, dopamine, endothelin, EGF, estrogen receptor, androgen receptor, GABA, glucocorticoid, glutamate, histamine, interleukin, melatonin, nicotinic acetylcholine, opiate, potassium channel, serotonin, and GABA transporter. The tested AUTOTAC and ATL did not significantly (above 50%) inhibit any of them. While the off-target specificity and selectivity of AUTOTAC/ATL require further confirmation, we believe that these results (in addition to non-interaction with NBR1) significantly validate the selectivity of AUTOTAC/ATL. Given that our current paper represents a proof-of-concept study in development of a targeted degradation platform based on p62-dependent macroautophagy, we hope to address off-target and selectivity issues (including the results of the radioligand binding assay above) in more detail for follow-up studies.

Comment 4. page 5, line 135. I understood your discussion on my previous specific comment 6. However, the importance of positive cooperativity is so far known only for PROTACs, and these backgrounds does not appear before this sentence. I feel that the sudden mention of “positive cooperativity” may confuse the reader. Note that in this revised manuscript, the author does not provide conclusive evidence that "positive cooperativity" is not important. There is only an example of a compound with a long linker, which does not convincingly argue for the general importance/non-importance of cooperativity in AUTOTAC experiments.

Reply: In our original manuscript, we showed that mutant p62 lacking its UBA domain could still bind mutant tau upon AUTOTAC treatment (Supplementary Fig. 6j). Thus, combining our interpretation of this finding with the fact that an AUTOTAC with a longer linker length still displayed efficient degradation, we felt that AUTOTACs may not be extremely reliant upon positive cooperativity as other degraders (e.g., PROTAC). However, we have now moved this interpretation to the Discussion sections to improve readability (lines 459-463).

Comment 5. page 6, line 154. Please provide the chemical structures of biotinylated ATLs in the figures.

Reply: The structures of the biotinylated ATLs are now included in **Supplementary Fig. 1e** (lines 145 and 155)

Comment 6. page 6, line 159. WIPI2 and LAMP1 localize at different steps in the autophagy processes. The colocalization data in Suppl Fig.2 are not quantified and do not provide any time course-related information. Overall, the revisions made to show the recruitment of autophagosome membrane to the p62 aggregates do not sufficiently address the concerns in my previous comments (major comments 1).

Reply: We quantified the co-localization of p62 with WIPI2 as punctate structures, indicative of autophagic membranes, after a 24 hour-treatment of ATL, now shown in **Supplementary Fig. 2i-j** (lines 157-162). We also performed a new set of experiments showing that WIPI2 colocalizes with p62 at least as early as 6 hours post-ATL treatment (**Supplementary Fig. 2k-l**; lines 157-162). Since WIPI2 localizes to omegasome-anchored phagophores and recruits the ATG cascade complex for eventual LC3 lipidation, we believe that this supports the recruitment of p62 to autophagic membranes upon ATL treatment.

Comment 7. page 7, lines 175-185. (Regarding my previous specific comments 12 and 13). These sentences are speculations without sufficient support of experimental evidence. I

would like to suggest removing these from the Result section and including them in the Discussion section.

Reply: The sentences in question have now been moved from the Result section to the Discussion section (lines 415-422).

Comment 8. page 9, line 229. The observation that the level of substrate did not increase 8 hours after AUTOTAC wash out is not reliable evidence to discuss whether AUTOTAC has catalytic properties.

Reply: We have now revised the sentence to read "...suggesting that AUTOTACs may display sustained degradative efficacy and be recycled from the lysosome" (lines 221-223). We have also revised all claims of catalytic efficacy to sustained efficacy in the revised manuscript.

Comment 9. page 11, line 282 and Suppl Fig. 5c. The levels of Atg7 protein were not sufficiently decreased by Atg7 siRNA treatment. This experiment is not trustworthy. The conclusion that AUTOTAC is autophagy-dependent cannot be validated by this data. Please also note that bands of LC3-II appear in the 2 lanes in the WB image where the author claimed the Atg7 knock-down. If Atg7-KD was successful, autophagy activity would decrease so that LC3-II bands are expected to disappear. To obtain conclusive evidence that AUTOTAC is indeed autophagy-dependent, I would like to suggest avoid using RNAi of Atg p7 and use Atg KO cell lines. The authors used Atg5 KO MEF cells in a Suppl Fig. Please replace all the experiments using Atg7 RNAi with the data using Atg5 KO cells.

Reply: We have experimentally observed that the disappearance of LC3-II bands, as astutely pointed out, occurs in a cell line- and siRNA type-dependent manner. Given that AUTOTAC-mediated degradation of FK2 was completely abolished by the ATG7 knockdown, and that other groups have also reported similar observations of a lack of LC3-II band disappearance upon ATG7 knockdown (depending on the siRNA used), we feel that our results are validated. However, to conclusively verify autophagy-dependent MoA of AUTOTACs, we performed a series of additional experiments using ATG5 wild-type and knock-out MEF cells transiently expressing mutant desmin. We observed that, similar to our ATG7 interference data, AUTOTAC-mediated degradation of mutant desmin observed in wild-type MEFs was abolished in ATG5 knock-out counterparts. The new data is shown as **Supplementary Fig. 5i** with appropriate changes to the text (lines 301-304).

Comment 10 (regarding my previous specific comment 31). Page 12, lines 297-314. I could not understand the author's argument about WT and mutant desmin well. In this paragraph, it is discussed that mutant desmin, which is prone to aggregation, can be degraded by AUTOTAC, but wt-desmin is not. I was a little confused because the authors also state in this paper that some soluble proteins such as MetAP2 can be degraded by AUTOTACs. ATL can aggregate p62, but do you mean this ability alone is not sufficient? Does this mean that the substrate must have aggregate-prone nature for successful AUTOTAC degradation?

Reply: We induced the degradation of soluble oncoproteins (e.g., MetAP2) using target-binding ligands (TBLs) largely specific to only those proteins, respectively. However, in the case of wild-type vs. mutant desmin, we used PBA and anle138b as TBLs, which respectively recognize mutant and oligomeric/aggregated protein species. Since only mutant desmin forms misfolded oligomer/aggregates (wild-type desmin is monomeric and soluble), the PBA-based

and anle138b-based AUTOTACs exhibited degradative efficacy selectively against mutant desmin. While p62-ligands can indeed induce autophagy, we found that the degradative efficacy of the AUTOTACs were superior against aggregation-prone proteins (which is natural due to TBL-contributed selectivity).

Comment 11, part 1. page 13, line 330 and line 339. Here, using just one compound, the authors conclude that linker length does not affect AUTOTAC-mediated degradation. This should be judged carefully. Without several compounds with different length linkers, no accurate conclusion can be drawn. The efficiency of degradation is also affected by the magnitude of the interaction between TBL and the substrate. If the affinity of the TBL-substrate is small, the cooperativity of p62 with the substrate in the ternary complex may play a role. I think the current description is a bit overstated.

Reply: Thank you for your comment. While we agree that linker length-independence of the degradative efficacy of AUTOTACs should be further confirmed using several other compounds with varying linker lengths, we feel that our data is sufficient to at least suggest that AUTOTACs may not be as dependent on linker length as other previously published degraders, especially those that require positive cooperativity (e.g., PROTAC). Thus, we have toned down the sentence in question to suggest, and not conclude, linker length-independence (lines 323-327). We wholeheartedly agree that the efficiency of degradation is affected by TBL-target interaction affinity/kinetics, especially for low-affinity TBL-target interactions. We have also included this point in our Discussion section (lines 462-463).

Comment 11, part 2. Page 13, line 339. As I wrote as specific comment 8, The observation that the level of mutant Tau did not increase 8 hours after wash-out of AUTOTAC is not fully reliable evidence to discuss whether AUTOTAC has catalytic properties. The revised sentence “These data validate catalytic efficacy of AUTOTACs” needs to be reconsidered.

Reply: As kindly suggested, we have now revised the sentence to read “... general linker length-insensitive and sustained efficacy...” (lines 332-333).

Comment 12 (regarding previous specific comment 38). Macroautophagy proceeds only in cytoplasm so that substrate with nuclear-localizing signal (NLS) are expected to escape from degradation but here you found the successful degradation of a protein with NLS. The authors' reply to my previous comment is below. “We speculate that the presence of nuclear subpopulations of p62 and LC3, which has been reported in the literature, allow AUTOTACs to target nucleus-resident mutant HttQ97 for autophagic degradation.” It does not make sense to me. Nuclear localization of p62 and LC3 does not show the formation of isolation membrane in nucleus, because the isolation membrane originates from ER-mitochondria contact site outside of nucleus. The result that proteins with NLS can be degraded makes me a little worried about the correctness of the mechanism depicted by the author in Fig. 7. There is no description of why AUTOTAC can degrade nuclear protein in the revised main text. Please include your comments in the main text.

Reply: As astutely pointed out, nuclear LC3 does not localize to autophagic membranes unless it is retrotranslocated and post-translationally modified (e.g., deacetylation/phosphorylation). We did not mean to say that AUTOTACs could target nucleus-resident mutant HttQ97 via formation of the phagophore *within the nucleus*. What we meant to say is that since nuclear p62 is known to retrotranslocate to the cytosol,

AUTOTACs will likely be able to bind nuclear p62 or cytosolic p62 that then translocates to the nucleus. The p62-AUTOTAC complex in the nucleus will then be able to 'drag' TBL-bound mutant HttQ97 to the cytosol via the nuclear pore complex (whose diameter can be large enough to fit p62 and/or HttQ97 oligomers/aggregates). We have revised the Result section in question to better address these possibilities (lines 376-380).

Comment 13. page 107 Fig. 7. Figure 7 contains both the results and speculations obtained by this study. It is difficult for the reader to distinguish between the two, and there is a concern that this may be misleading. For example, that AUTOTAC is recycled is a hypothesis, not directly proven by the data in this paper. As already pointed out, the wording p62 ligand should be used instead of ATL. Proof of substrate transfer to the autolysosome can be examined with substrates that have pH-sensitive fluorescence, but the data in Suppl. Fig. 6h of this manuscript is of insufficient quality. Changing the title of the figure to something like the Speculative model of AUTOTAC mechanism could be considered. It is also difficult to understand from Fig. 7 how preexisted intracellular aggregates are degraded by the AUTOTAC mechanism. Could you draw a model in which p62 is newly recruited by the action of AUTOTAC around aggregates such as Tau? Is it possible to do additional experiments to support such a hypothetical diagram?

Reply: As kindly pointed out, we have now revised Fig. 7 to better reflect both the results and speculations from our study. Specifically, the title of Fig. 7 has now been changed to "Speculative model of AUTOTAC and its mechanism-of-action" (line 1744). Also, we agree that AUTOTACs being recycled are not entirely proven in our paper. As such, in our original manuscript, we had emphasized this point with a dashed arrow (as opposed to a solid arrow). However, to better reflect this, we have changed "AUTOTAC is recycled" to "sustained efficacy (likely recycled)." Finally, we have changed "autophagy-targeting ligand (ATL)" to "p62-targeting ligand (ATL)" within the graphical illustration.

Regarding your point on the targeting of pre-existing aggregates with AUTOTACs, please refer to our answer on major comment #3. We feel that since our AUTOTACs display statistically significant degradative efficacy against pre-formed aggregates, our current graphical model accurately represents the mechanism-of-action of AUTOTACs.

Comment 14. page 109, Suppl. Fig. 2. Chemical structure of biotinylated YOK-2204 should be included. The title of the figure starts with "Computational modeling of" but the figure include data unrelated to the computational studies.

Reply: As recommended, we have now included all the chemical structures of biotinylated p62-targeting ligands in **Supplementary Fig. 1e** (lines 145 and 155). Additionally, Supplementary Fig. 2 now starts with "Characterization of..." (line 1752).

Comment 15. page 111 legends for b and c. "ATG-/-" Is it ATG5-/- MEF cells?

Reply: We apologize for the typo, which has been corrected (line 1771).

REVIEWER 3

Remarks to the Author: The authors have addressed all the criticisms raised by this reviewer.

REVIEWERS' COMMENTS

Reviewer #2 (Remarks to the Author):

Major Comments

Comment 1. (Regarding my previous comment 1)

The first author and contact author are members of AUTOTAC Bio Inc. and therefore, a clear distinction should be made between business and science.

The authors' assertions, i.e., their use of the term in U.S. patent applications and trademark applications, do not usually warrant preemption in academic papers. These IP documents are not peer-reviewed.

In academic papers, Autophagy-targeting chimera (AUTACs) is widely accepted as the formal name for different technologies, and the authors' manuscript should avoid these terms to avoid unnecessary confusion. This manuscript should be revised.

I recommend that the authors choose another appropriate name, but if the authors insist on using the name AUTOTAC in this paper, the following revision might be possible.

Remove the phrase "autophagy-targeting chimera" from the manuscript as used in the title and abstract of the paper, and drop the claim that AUTOTAC is an abbreviation for "autophagy-targeting chimera".

Specific Comments

Comment 1. (Regarding my previous specific comment 3)

The current manuscript does not provide sufficient data on off-target or selectivity issues. The authors appear to have additional experimental data that they would prefer not to include in this paper. I respect this decision.

For this reason, the authors should clearly state in the manuscript that the "off-target and selectivity issues" are not investigated in this paper.

In particular, I would like to ask for a note to be added to the discussion section.

Comment 2. (Regarding my previous specific comment 6)

As shown here for WIPI2 and p62, quantify the co-localization ratio for panel H (LAMP1) in Suppl. Fig. 2.

There are many errors in the legend of Suppl. Fig. 2. For example, there are two (j). The latter will probably be corrected as (i). (k) is described as a quantification of the data in (j), but (k) is not a graph.

(l) is stated to be a study of autophagy flux, so it will probably be corrected to (m) or (n).

(k) at the end should be corrected to (o).

Comment 3 (regarding my previous comment 8)

The authors have agreed with me not to state definitely that AUTOTAC has "catalytic properties". More experiments are needed to support this conclusion. Instead, the authors decided to use the term "sustained efficacy".

I believe that the wording in the following part of the manuscript may mislead people into thinking that catalytic properties are a certainly proven fact.

Please consider revising them.

line 225

“Taken together, these results suggest that AUTOTAC does not require ubiquitin-dependent and PPI-driven cooperativity for its catalytic autophagic proteolysis.”

line 478

“Also, given the catalytic nature of AUTOTACs”

line 471

“catalytic nature of degradation.”

Comment 4 (Regarding my previous comment 11)

The authors did not examine a series of compounds with varying linker lengths.

Please reconsider revising the following sentence in line 333, as no general conclusions could be drawn from their experiments.

“These data validate the general linker length-insensitive”

Comment 5

In the discussion section, please focus on what has been reliably proven from the experimental results, the relationship with previous studies, and what needs further study (e.g., off-target and selectivity of ATL were not sufficiently studied in this study).

It is fine to mention the mechanism by inference in Fig. 7, but please make a clear distinction between what has been proven and what is inference.

As I pointed out in an earlier comment, the idea that AUTOTAC acts catalytically remains a hypothesis.

REVIEWER 2

Major comments

Comment 1. (Regarding my previous comment 1)

The first author and contact author are members of AUTOTAC Bio Inc. and therefore, a clear distinction should be made between business and science. The authors' assertions, i.e., their use of the term in U.S. patent applications and trademark applications, do not usually warrant preemption in academic papers. These IP documents are not peer-reviewed. In academic papers, Autophagy-targeting chimera (AUTACs) is widely accepted as the formal name for different technologies, and the authors' manuscript should avoid these terms to avoid unnecessary confusion. This manuscript should be revised. I recommend that the authors choose another appropriate name, but if the authors insist on using the name AUTOTAC in this paper, the following revision might be possible. Remove the phrase "autophagy-targeting chimera" from the manuscript as used in the title and abstract of the paper, and drop the claim that AUTOTAC is an abbreviation for "autophagy-targeting chimera".

Reply: As astutely pointed out, the current affiliations of the first and the contact author do include AUTOTAC Bio Inc. (business/industry); however, they also include Seoul National University (academia/science), from which the proof-of-concept (and the vast majority of the data shown in this manuscript) of the AUTOTAC platform was first proposed and established. We recognize that the phrase “autophagy-targeting chimera” may be already introduced as AUTAC in the scientific community. As such, when we refer to AUTOTAC in its full, non-abbreviated term (only found in the Abstract section), we have adopted the following: “AUTOPhagy-TARgeting Chimera.”

Specific comments

Comment 1. (Regarding my previous specific comment 3).

The current manuscript does not provide sufficient data on off-target or selectivity issues. The authors appear to have additional experimental data that they would prefer not to include in this paper. I respect this decision. For this reason, the authors should clearly state in the manuscript that the "off-target and selectivity issues" are not investigated in this paper. In particular, I would like to ask for a note to be added to the discussion section.

Reply: As recommended, we have stated that “...the off-target and selectivity issues have yet to be fully investigated and should be addressed in follow-up studies” in the Discussion section (lines 477-478).

Comment 2. (Regarding my previous specific comment 6).

As shown here for WIPI2 and p62, quantify the co-localization ratio for panel H (LAMP1) in Suppl. Fig. 2. There are many errors in the legend of Suppl. Fig. 2. For example, there are two (j). The latter will probably be corrected as (i). (k) is described as a quantification of the data in (j), but (k) is not a graph. (l) is stated to be a study of autophagy flux, so it will probably be corrected to (m) or (n). (k) at the end should be corrected to (o).

Reply: We apologize for the typos and mislabeling in the figure legends of Supplementary Figure 2, and have corrected them. Additionally, the co-localization ratio between LAMP1 and p62 has been quantified and is now shown as **Suppl. Fig. 2i**.

Comment 3. (Regarding my previous comment 8).

The authors have agreed with me not to state definitely that AUTOTAC has "catalytic properties". More experiments are needed to support this conclusion. Instead, the authors decided to use the term "sustained efficacy". I believe that the wording in the following part of the manuscript may mislead people into thinking that catalytic properties are a certainly proven fact. Please consider revising them.

line 225

“Taken together, these results suggest that AUTOTAC does not require ubiquitin-dependent and PPI-driven cooperativity for its catalytic autophagic proteolysis.”

line 478

“Also, given the catalytic nature of AUTOTACs”

line 471

“catalytic nature of degradation.”

Reply: As kindly pointed out, all mentions of “catalytic efficacy” have been revised to “sustained efficacy” (lines 227, 438).

Comment 4. (Regarding my previous comment 11).

The authors did not examine a series of compounds with varying linker lengths. Please reconsider revising the following sentence in line 333, as no general conclusions could be drawn from their experiments. “These data validate the general linker length-insensitive”

Reply: We have rephrased the sentence in question to now read “These data raise the likelihood that AUTOTACs are linker length-insensitive and exhibit sustained efficacy against tau oligomers and aggregates” (lines 334-335).

Comment 5. In the discussion section, please focus on what has been reliably proven from the experimental results, the relationship with previous studies, and what needs further study (e.g., off-target and selectivity of ATL were not sufficiently studied in this study). It is fine to mention the mechanism by inference in Fig. 7, but please make a clear distinction between what has been proven and what is inference. As I pointed out in an earlier comment, the idea that AUTOTAC acts catalytically remains a hypothesis.

Reply: The Discussion section has now been revised to clearly distinguish among conclusions drawn from experimental results, the relationship with previous studies, and follow-up studies. Additionally, references to Figure 7 within either the Result or the Discussion section have been toned down in nuance with respect to their speculative nature.